# SEMANTIC FLOW: LEARNING SEMANTIC FIELDS OF DYNAMIC SCENES FROM MONOCULAR VIDEOS

**Fengrui Tian**[1]    **Yueqi Duan**[2]    **Angtian Wang**[3]    **Jianfei Guo**[4]    **Shaoyi Du**[1*]
[1]National Key Laboratory of Human-Machine Hybrid Augmented Intelligence,
National Engineering Research Center for Visual Information and Applications,
and Institute of Artificial Intelligence and Robotics, Xi'an Jiaotong University
[2]Tsinghua University    [3]Johns Hopkins University    [4]Shanghai AI Laboratory
`tianfr@stu.xjtu.edu.cn`   `duanyueqi@tsinghua.edu.cn`
`angtianwang@jhu.edu`   `guojianfei@pjlab.org.cn`
`dushaoyi@xjtu.edu.cn`

## ABSTRACT

In this work, we pioneer Semantic Flow, a neural semantic representation of dynamic scenes from monocular videos. In contrast to previous NeRF methods that reconstruct dynamic scenes from the colors and volume densities of individual points, Semantic Flow learns semantics from continuous flows that contain rich 3D motion information. As there is 2D-to-3D ambiguity problem in the viewing direction when extracting 3D flow features from 2D video frames, we consider the volume densities as opacity priors that describe the contributions of flow features to the semantics on the frames. More specifically, we first learn a flow network to predict flows in the dynamic scene, and propose a flow feature aggregation module to extract flow features from video frames. Then, we propose a flow attention module to extract motion information from flow features, which is followed by a semantic network to output semantic logits of flows. We integrate the logits with volume densities in the viewing direction to supervise the flow features with semantic labels on video frames. Experimental results show that our model is able to learn from multiple dynamic scenes and supports a series of new tasks such as instance-level scene editing, semantic completions, dynamic scene tracking and semantic adaption on novel scenes.

## 1 INTRODUCTION

In recent years, Neural Radiance Field (NeRF) (Mildenhall et al., 2020) methods bring a storm on scene reconstruction tasks (Mildenhall et al., 2022; Chen et al., 2021; Yu et al., 2021; Niemeyer & Geiger, 2021; Deng et al., 2022). NeRF represents the static scene as an implicit neural network that takes an input of the position and viewing direction at a point in the 3D space, and outputs the corresponding color and density. The high reconstruction quality facilitates many researchers to extend its applications toward dynamic scenes (Gao et al., 2021; Wu et al., 2022; Park et al., 2021; Weng et al., 2022), which are mostly recorded by monocular cameras in the real world.

Dynamic scene reconstruction from monocular videos is a challenging problem. Since the foreground is dynamically changing in the scene, learning dynamic radiance fields from monocular videos suffers from 2D-to-3D ambiguity in the viewing direction when mapping the observed object motions on the 2D images (*i.e.,* optical flows) to dense object motions in the 3D scene (*i.e.,* scene flows (Vedula et al., 1999)). Previous works mainly address the challenge and render the scene by designing spatio-temporal constraints (Peng et al., 2021; Qiao et al., 2022; Li et al., 2021; Gao et al., 2021; Tian et al., 2023). Specifically, they assume that points are temporally moving in the scene, and reconstruct the scene with consistency constraints along point trajectories.

While the aforementioned dynamic NeRF methods successfully build radiance fields for dynamic scenes, the semantics in these fields remain underexplored. Learning the semantic information in

---

[*]Corresponding author. Codes are available at `https://github.com/tianfr/Semantic-Flow/`.

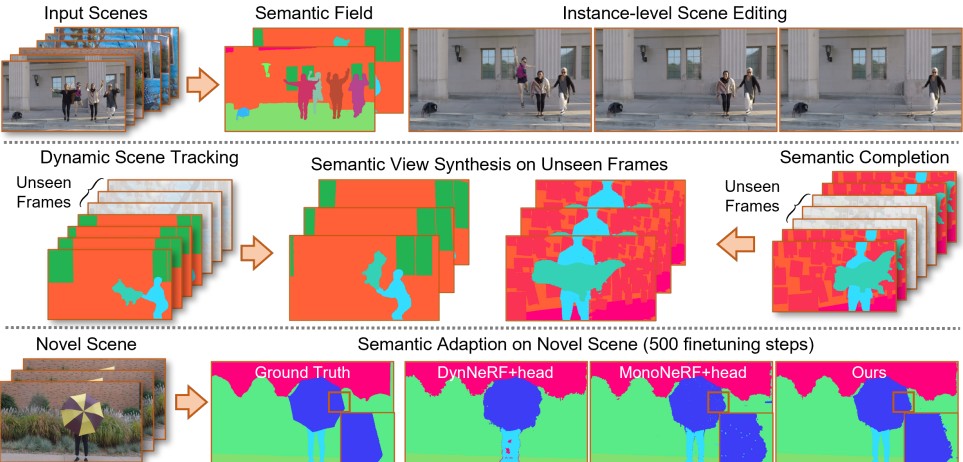

Figure 1: Semantic Flow learns from flows capturing motion information in dynamic scenes. In this way, Semantic Flow can learn semantics from multiple scenes and conduct instance-level editing (**Top**). It also supports dynamic scene tracking and semantic completion (**Middle**), both of which learn with few semantic labels. Compared to DynNeRF (Gao et al., 2021) and MonoNeRF (Tian et al., 2023), Semantic Flow can transfer to novel scenes with more accurate details (**Bottom**).

dynamic scenes is considered beneficial since such information could support the further interpretability and applications of the reconstructed scenes. The potential information in the flow of each moving object is important for many computer vision tasks such as trajectory prediction (Xu et al., 2023), action recognition (Feichtenhofer et al., 2019) and other applications (Trucco & Plakas, 2006; Tsai et al., 2016; Alahi et al., 2016; Hu et al., 2023).

To learn semantics in dynamic radiance fields, an intuitive solution is to add a semantic segmentation head to the previous dynamic NeRF methods. In this way, the semantics of each point is estimated from the position, timestamp, and other information related to the point. Although such approach is sufficient for learning color and density, it cannot provide the motion information related to the scene. In other words, the motion of an object can be observed from its continuous flow over time rather than a single point at each moment. Due to the lack of motion information, predicting semantics from points forces the model to overfit to the training views in the current scene, which limits its generalization performance when training with few annotated labels or transferring to novel scenes. Such limitation heavily restrains the model from a wider range of applications in the world.

In this paper, we propose a neural semantic representation for dynamic scenes. Instead of learning from separate points, we propose to learn from continuous flows that capture motion information of scenes. While extracting 3D flow features from 2D frames suffers from 2D-to-3D ambiguity problem, we consider the volume densities as opacity priors describing the contributions of flow features to the semantic labels on frames, so that the 2D semantic labels could be used to supervise the semantic field by integrating the flow features in the viewing direction.

To achieve this, we propose Semantic Flow for building semantic fields of dynamic scenes from flows. We first design a flow network to predict flows in the dynamic scene. Then, we exploit the locations of the points on the flows as indexes to extract local image features from video frames, and aggregate the extracted local features and flow displacements as flow features. We design a flow attention module to fully exploit motion information from flows and employ a semantic network to output the semantic logits of each flow. We aggregate the semantic logits in the viewing direction with volume densities and supervise the logits with semantic labels on the video frames.

In addition, we also develop a dataset called Semantic Dynamic Scene Dataset based on the dynamic scenes from the Dynamic Scenes dataset (Yoon et al., 2020) to conduct semantic tasks. In our dataset, there are seven forward-facing dynamic scenes with complex foreground motions. We manually annotate the pixel-wise semantic labels of each video, and conduct a series of experiments on the dataset. As shown in Figure 1, Semantic Flow can learn from multiple dynamic scenes, and

present strong generalization ability on various tasks such as semantic adaption on novel scenes, semantic completion, and dynamic scene tracking.

## 2    RELATED WORK

**NeRF for semantics.** In recent years, NeRF achieves great progress on the novel view synthesis task by representing scenes as neural implicit representations (Mildenhall et al., 2020; Liu et al., 2020; Gafni et al., 2021; Yu et al., 2021; Niemeyer & Geiger, 2021; Gu et al., 2022; Zhi et al., 2021; Martin-Brualla et al., 2021; Srinivasan et al., 2021). Such success in scene reconstruction tasks encourages many researchers to explore its possibility in scene understanding tasks. The first attempt is the Semantic NeRF (Zhi et al., 2021), which exploits many semantic tasks in the neural radiance field by adding a semantic head to the origin NeRF pipeline. Based on that, recent works focus on the semantic understanding of static scenes (Yang et al., 2021; Liu et al., 2023a; Vora et al., 2022; Fan et al., 2023; Kundu et al., 2022). Liu et al. (2023a) studied the generalization ability of semantic radiance field by introducing semantics into each ray. Vora et al. (2022) proposed NeSF that is able to reason semantics from geometry stored in the radiance field. Fan et al. (2023) proposed NeRF-SOS that studies the segmentation problem in a self-supervised way. Although above mentioned methods achieve promising performance, semantic learning in dynamic scenes remains underexplored.

**Dynamic NeRF from monocular videos.** The monocular videos are commonly captured by personal phones in daily life. Hence many researchers make huge progress on building the dynamic radiance field from monocular videos (Li et al., 2021; Gao et al., 2021; Park et al., 2021; Pumarola et al., 2020; Peng et al., 2021; Qiao et al., 2022; Ost et al., 2021; Weng et al., 2022; Gafni et al., 2021). The major challenge behind scene reconstruction from monocular videos is the ambiguity problem, which means the same observed image sequences can be inferred from different scene reconstructions. Initially, some works extend the vanilla NeRF to dynamic scenes by introducing a time dimension (Li et al., 2021; Gao et al., 2021; Pumarola et al., 2020). Based on that, a series of studies try to reconstruct the dynamic parts of the scene more precisely with shape priors (Peng et al., 2021; Qiao et al., 2022; Ost et al., 2021; Weng et al., 2022; Gafni et al., 2021). Besides, Tian et al. (2023) studied the generalization problem and proposed MonoNeRF which builds a generalizable dynamic radiance field by jointly optimizing the spatial and temporal features. In this study, we explore the dynamic radiance field with semantic representations.

## 3    SEMANTIC FLOW

In this section, we introduce our model which we term as Semantic Flow. Given a monocular video $I$ containing $N$ frames $\{I_1, I_2, ..., I_N\}$ with known camera poses $\{P_1, P_2, ..., P_N\}$, we denote $I_t, P_t$ as the frame and pose at timestamp $t$ and hence $t \in \{1, 2, ..., N\}$. We use semantic labels to delineate foregrounds in each video frame, and reconstruct semantic fields of the dynamic foreground and static background separately. We follow previous works (Li et al., 2021; Gao et al., 2021; Tian et al., 2023) and supervise the model with optical flow and depth signals, which can be readily obtained from the state-of-the-art pretrained models (Teed & Deng, 2020; Ranftl et al., 2022). For the semantic field of dynamic foreground, we first build a flow field based on the input video to calculate the flow of each point in the scene. Based on that, we exploit the moving points on the flows as indexes to locate local image patches on each frame and extract the local image features. We combine the local features and flow displacements as flow features and design a flow attention module to uncover the semantic information related to the motion information of flows. To supervise the semantic field with labels on video frames, we employ a semantic network to output semantic logits of each flow, and render the logits of each pixel on the frames by integrating the semantic logits of flows in the viewing direction with volume densities. For the semantic field of static background, we directly sample the semantic features of each point from video frames. Our pipeline is shown in Figure 2. Before introducing our model, we first present a review of concurrent dynamic radiance fields.

### 3.1    PRELIMINARIES

To build a neural radiance field for dynamic scenes, previous works (Gao et al., 2021; Li et al., 2021; Tian et al., 2023) build static and dynamic radiance fields for reconstructing backgrounds

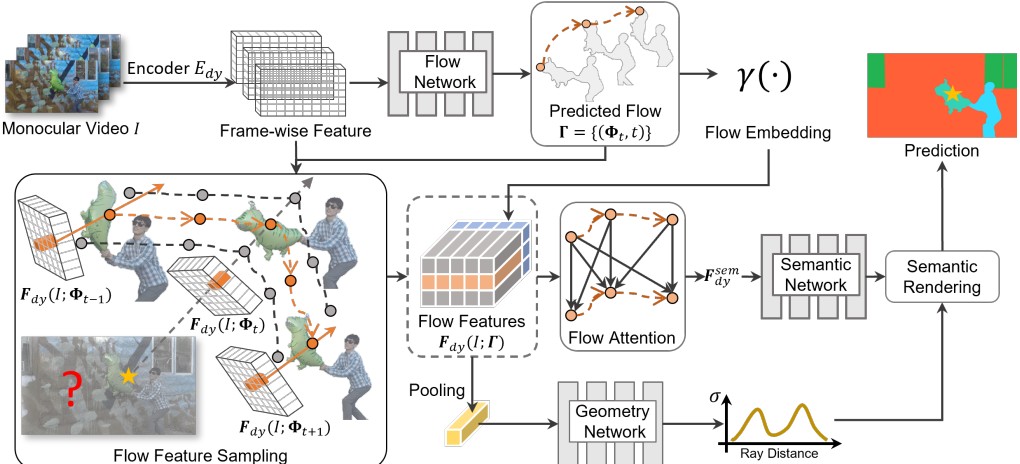

Figure 2: **The overview of the proposed model**. We first design a flow network to predict flows in the dynamic scene. Then, taking the orange flow (in the bottom left part) as an example, we aggregate the flow features from the feature map of each frame. We propose the flow attention module to reveal the motion information from the aggregated flow features. Finally, we design a semantic network to output the semantic logits of each flow, and predict semantics on the frame by rendering the semantic logits along camera rays with volume densities $\sigma_{dy}$ as opacity priors.

and foregrounds separately and blend static and dynamic radiance fields to render the entire scene with learned blending weights. Specifically, for static radiance field, they learn an implicit function $\Psi_{st} : (\sigma_{st}, \boldsymbol{c}_{st}) = \Psi_{st}(\boldsymbol{x})$ that converts the input coordinate $\boldsymbol{x} \in \mathbb{R}^3$ to the corresponding color $\boldsymbol{c}_{st} \in \mathbb{R}^3$ and density $\sigma_{st} \in \mathbb{R}$. Then they exploit the volume rendering technology to synthesize novel view images of static backgrounds. Given a ray $\boldsymbol{r} = \boldsymbol{o} + u\boldsymbol{d}$ emitting from the camera center $\boldsymbol{o} \in \mathbb{R}^3$ along the direction $\boldsymbol{d} \in \mathbb{R}^3$ through a pixel on the image, the volume rendering function can be formulated as

$$\boldsymbol{C}_{st}(\boldsymbol{r}) = \int_{u_n}^{u_f} T_{st}(u)\sigma_{st}(u)\boldsymbol{c}_{st}(u)du, \tag{1}$$

wherein $u_n, u_f$ are the upper and lower bounds of the rendering depth range and $T_{st}(u) = \exp(-\int_{u_n}^{u} \sigma_{st}(s)ds)$ is the accumulated transmittance. We simplify $\boldsymbol{c}(u) = \boldsymbol{c}(\boldsymbol{r}(u))$ and $\sigma(u) = \sigma(\boldsymbol{r}(u))$. For the dynamic radiance field, it takes the position $\boldsymbol{x}$ and time $t$ as input and learns an implicit function $\Psi_{dy} : (\sigma_{st}, \boldsymbol{c}_{st}, b) = \Psi_{dy}(\boldsymbol{x}, t)$, where $b$ is the blending weight that decides whether the point belongs to the dynamic foreground (Tian et al., 2023). Similarly, novel view images of dynamic foreground could be also synthesized by (1). The static and dynamic radiance fields $\Psi_{st}, \Psi_{dy}$ collaboratively render the novel view images of the entire scene,

$$\boldsymbol{C}_{full}(\boldsymbol{r}) = \int_{u_n}^{u_f} T_{full}(u)\sigma_{full}(u)\boldsymbol{c}_{full}(u)du, \tag{2}$$

where $\sigma_{full}(u)\boldsymbol{c}_{full}(u) = (1-b)\sigma_{st}(u)\boldsymbol{c}_{st}(u) + b\sigma_{dy}(u)\boldsymbol{c}_{dy}(u)$. In this paper, we follow the principle to separately reconstruct the semantic fields of dynamic foreground and static background.

## 3.2 SEMANTIC FIELD FOR DYNAMIC FOREGROUND

In this section, we introduce our semantic field for dynamic foreground. We first describe our implicit flow field which generates the flows in the dynamic scenes. Then, we introduce the flow feature aggregation module and flow attention module to predict semantics of each flow. In the end, we present our semantic rendering method.

**Implicit flow field.** We build a flow field to trace the flow on each point. Concretely, we first extract the video features $\boldsymbol{F}_{dy} = E_{dy}(I)$ by a video encoder $E_{dy}$. Then, we build a flow field with a multi-layer perceptron (MLP) $\Psi_{flow}$ based on $\boldsymbol{F}_{dy}$. Given a point position $\boldsymbol{x}$ at timestamp $t$, the flowing point positions $\boldsymbol{\Phi}_{t-1}, \boldsymbol{\Phi}_{t+1} \in \mathbb{R}^3$ at timestamps $t-1, t+1$ can be calculated as

$$(\boldsymbol{\Phi}_{t-1}, \boldsymbol{\Phi}_{t+1}) = \Psi_{flow}(\boldsymbol{F}_{dy}; \boldsymbol{\Phi}_t, t), \tag{3}$$

where $\mathbf{\Phi}_t = \boldsymbol{x}$. Then, the point trajectories $\mathbf{\Phi}_{t-r}, \mathbf{\Phi}_{t+r}$ at timestamps $t-r, t+r$ could be inferred from (3) by introducing $\mathbf{\Phi}_{t-r+1}, \mathbf{\Phi}_{t+r-1}$ and $t-r+1, t+r-1$ into $\Psi_{flow}$. In this way, we can derive the flow related to a point as a set of point positions with timestamps $\mathbf{\Gamma} = \{\mathbf{\Gamma}_1, \mathbf{\Gamma}_2, ..., \mathbf{\Gamma}_N\}$ where $\mathbf{\Gamma}_t = (\mathbf{\Phi}_t, t)$. We follow previous works (Tian et al., 2023; Gao et al., 2021; Li et al., 2021) and employ optical flows as the supervision signal. Concretely, for a point $\boldsymbol{x} = \boldsymbol{r}(u)$ on the ray at timestamp $t$, we obtain its flow with $\Psi_{flow}$. We estimate the optical flow by integrating the flow displacements of the points on the ray with densities $\sigma_{dy}$. The estimated flows are then supervised by the flows generated from the pretrained model (Teed & Deng, 2020). In the following, we build the flow features based on the flow of each point.

**Flow feature aggregation.** Given a point with its position $\boldsymbol{x}$, timestamp $t$ and flow $\mathbf{\Gamma}$, we sample the flow features from video features over the flow of the point. Specifically, for each $\mathbf{\Gamma}_t = (\mathbf{\Phi}_t, t)$, we employ the camera pose $\boldsymbol{P}_t$ to project the $\mathbf{\Phi}_t$ onto the video frame $I_t$ as $\boldsymbol{P}_t(\mathbf{\Phi}_t)$. We extract the feature map of $I_t$ by $E_{dy}$ and exploit $\boldsymbol{P}_t(\mathbf{\Phi}_t)$ as index to sample the point feature vector on the feature map with bilinear interpolation,

$$\boldsymbol{F}_{dy}(I; \mathbf{\Phi}_t) = \Omega(E_{dy}(I_t); \boldsymbol{P}_t(\mathbf{\Phi}_t)), \tag{4}$$

where the function $\Omega(\cdot)$ is the bilinear interpolation. $E_{dy}(I_t)$ denotes the frame-wise feature map extracted by $E_{dy}$. The flow features vector at any timestamp $\tau$ could be built based on the point feature vector with the relative flow displacement $\Delta\mathbf{\Gamma} = \mathbf{\Gamma}_\tau - \mathbf{\Gamma}_t$,

$$\boldsymbol{F}_{dy}(I; \mathbf{\Gamma}_\tau) = \{\boldsymbol{F}_{dy}(I; \mathbf{\Phi}_\tau), \gamma(\Delta\mathbf{\Gamma})\}, \tag{5}$$

where the function $\gamma(\cdot)$ denotes the position embedding function. The initial flow features are defined as

$$\boldsymbol{F}_{dy}(I; \mathbf{\Gamma}) = [\boldsymbol{F}_{dy}(I; \mathbf{\Gamma}_1), \boldsymbol{F}_{dy}(I; \mathbf{\Gamma}_2), ..., \boldsymbol{F}_{dy}(I; \mathbf{\Gamma}_N)]^T. \tag{6}$$

In the following, we introduce our flow attention module for uncovering the motion information of the flow.

**Flow attention.** As the flow features sampled from video features provide specific information for object motions from different views and timestamps, we propose a flow attention module to reason the semantic features related to flow motions. Concretely, for each flow feature $\boldsymbol{F}_{dy}(I; \mathbf{\Gamma})$, our flow attention module is defined as

$$Q = \boldsymbol{F}_{dy}(I; \mathbf{\Gamma}) \times W_q, K = \boldsymbol{F}_{dy}(I; \mathbf{\Gamma}) \times W_k, V = \boldsymbol{F}_{dy}(I; \mathbf{\Gamma}) \times W_v,$$

$$A^{(h)} = \text{softmax}(\frac{Q^{(h)}K^{(h)T}}{\sqrt{d_k}}V^{(h)}), h \in \{1, ..., H\}, \tag{7}$$

where $d_k = C/H$ is the dimension of each head. $Q, K, V \in \mathbb{R}^{C \times C}$ denote the query, key and value features extracted from $\boldsymbol{F}_{dy}(I; \mathbf{\Gamma})$ by the fully connected layers $W_q, W_k, W_v$. We employ a multi-head attention module and $Q = [Q^{(1)}, Q^{(2)}, ..., Q^{(H)}], K = [K^{(1)}, K^{(2)}, ..., K^{(H)}], V = [V^{(1)}, V^{(2)}, ..., V^{(H)}]$ where $Q^{(h)}, K^{(h)}, V^{(h)} \in \mathbb{R}^{N \times d_k}$. $A^{(h)}$ is the output from $h$-th head. We exploit a semantic network implemented by a MLP $\Psi_o$ to obtain the final semantic feature vector from the outputs of all heads $\{A^{(1)}, A^{(2)}, ..., A^{(N)}\}$,

$$\boldsymbol{F}_{dy}^{sem}(I; \mathbf{\Gamma}) = \Psi_o(A^{(1)}, A^{(2)}, ..., A^{(N)}), \tag{8}$$

wherein $\boldsymbol{F}_{dy}^{sem}(I; \mathbf{\Gamma})$ denotes the final semantic feature vector of the flow.

**Semantic rendering**. As there is ambiguity in the viewing direction when estimating precise semantic labels in 3D space from the semantic labels on 2D video frames, we propose to supervise the semantic field by aggregating the semantics along the camera ray with the corresponding volume densities as opacity priors. Concretely, we design a semantic network $\Psi_{dy}^{sem} : \boldsymbol{s}_{dy}(\mathbf{\Gamma}) = \Psi_{dy}^{sem}(\boldsymbol{F}_{dy}^{sem}(I; \mathbf{\Gamma}))$ implemented by an MLP to generate the semantic logits $\boldsymbol{s}_{dy}(\mathbf{\Gamma})$ of the flow. Given a ray $\boldsymbol{r}$ starting from the camera center through a pixel on the image, for each point $\boldsymbol{r}(u)$ on the camera ray, we find its flow in the scene and calculate the corresponding semantic logits of the flow. The semantic logits of the ray are calculated by the following integration,

$$\boldsymbol{s}_{dy}(\boldsymbol{r}) = \int_{u_n}^{u_f} T_{dy}(u)\sigma_{dy}(u)\boldsymbol{s}_{dy}(\mathbf{\Gamma}(u))du, \tag{9}$$

where $s_{dy}(\Gamma(u))$ denotes the semantic logits of point $r(u)$ obtained from it flow feature vector. We simplify $s_{dy}(\Gamma(u)) = s_{dy}(\Gamma(r(u)))$ here. $\sigma_{dy}$ is predicted by our geometric network $\Psi_{geo}$. We supervise the rendered labels with ground truths by employing a multi-class crossentropy loss:

$$\mathcal{L}_{dy}^{sem}(r) = -\sum_{r} \Big[\sum_{l=1}^{L} p^l(r)\log\hat{p}_{dy}^l(r)\Big], \tag{10}$$

wherein $\hat{p}_{dy}^l, p^l$ are the semantic probabilities of class $l$ from the prediction and ground truth map, respectively. $L$ is the total number of classes and $1 \le l \le L$.

## 3.3 SEMANTIC FIELD FOR STATIC BACKGROUND

In this section, we introduce our semantic field for the static background. Since some parts of the background may be occluded by the changing foreground, the semantic features extracted from video frames in these parts always imply the foreground information. To obtain the correct semantic features of occluded background parts, we choose to extract the features from non-occluded views as the following equation,

$$F_{st}(I; x) = \Omega\big(E_{st}(I^*; P^*(x))\big), \tag{11}$$

where $E_{st}$ is the encoder of the static scene. $I^*, P^*$ denote the non-occluded frame and corresponding camera pose. Following Tian et al. (2023), we employ the straightforward *random* sampling strategy by effectively choosing one frame in the video. Then, the semantic logits in the static scene is represented by an MLP $\Psi_{st}^{sem} : s_{st}(x) = \Psi_{st}^{sem}(F_{st}(I; x))$. Similar to the semantic field for dynamic foreground described in Section 3.2, for each point $r(u)$ on the ray, we exploit volume densities to render the semantic logits of the ray in the static scene,

$$s_{st}(r) = \int_{u_n}^{u_f} T_{st}(u)\sigma_{st}(u)s_{st}(r(u))du, \tag{12}$$

where $s_{st}(r(u))$ denotes the semantic logits of the point $r(u)$. We supervise the logits by crossentropy loss:

$$\mathcal{L}_{st}^{sem}(r) = -\sum_{r} \Big[\sum_{l=1}^{L} p^l(r)\log\hat{p}_{st}^l(r)\Big], \tag{13}$$

where $\hat{p}_{st}^l$ is the predicted semantic probability of $l$-th class in the static scene.

## 3.4 OPTIMIZATION

During the training phrase, we first pretrain the semantic field of the static background by using the semantic labels that belong to the background, and then optimize the semantic field of the dynamic foreground combined with the pretrained static background field to render the entire semantic field,

$$s_{full}^{sem}(r) = \int_{u_n}^{u_f} T_{full}(u)\sigma_{full}(u)s_{full}^{sem}(r(u))du, \tag{14}$$

where $\sigma_{full}(u)s_{full}^{sem}(u) = (1-b)\sigma_{st}(u)s_{st}^{sem}(r(u)) + b\sigma_{dy}(u)s_{dy}^{sem}(\Gamma(u))$. $b$ denotes whether the point belongs to the dynamic foreground or static background. The semantic loss of the entire scene is defined as

$$L_{full}^{sem}(r) = -\sum_{r} \Big[\sum_{l=1}^{L} p^l(r)\log\hat{p}_{full}^l(r)\Big], \tag{15}$$

where $\hat{p}_{full}^l$ is the predicted semantic probability of class $l$. In the following, we apply several optimization strategies to the model for rendering the semantic field with better quality.

**Semantic consistency constraint**. We suppose that the semantics of flows are consistent over time across the whole dynamic scene. Specifically, $r_\tau$ denotes the warped ray $r$ at timestamp $\tau$ by employing the flows of the points on the ray, *i.e.*, $r_\tau(u) = \Phi_\tau(r(u))$, where $\Phi_\tau(r(u))$ denotes the new position of $r(u)$ when moving from the original timestamp to timestamp $\tau$. We supervise the semantic label of $r_\tau$ with the ground truth label of the ray $r$,

$$\mathcal{L}_{consis} = -\sum_{r_\tau} \Big[\sum_{l=1}^{L} p^l(r)\log\hat{p}_{dy}^l(r_\tau)\Big], \tag{16}$$

where $\hat{p}_{dy}^l(r_\tau)$ is $l$-th class semantic possibility of the ray $r_\tau$.

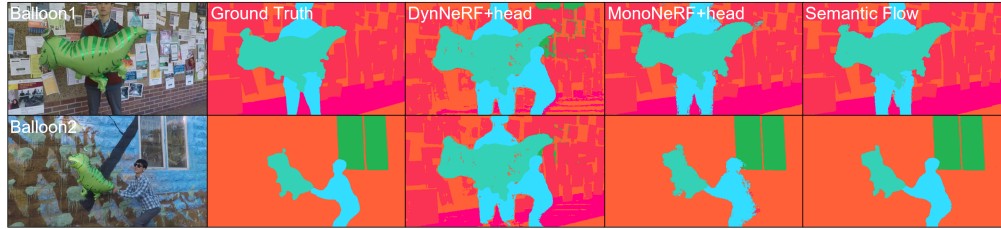

(a) Learning semantics from multiple dynamic scenes.

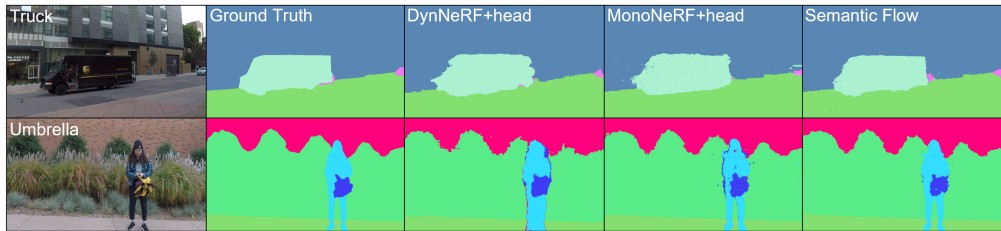

(b) Semantic adaption on novel scenes (500 finetuning steps).

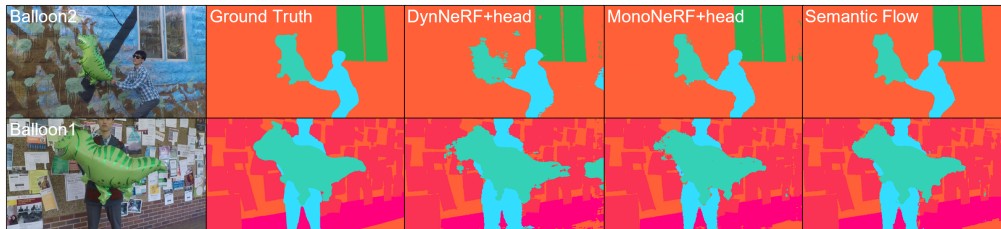

(c) Semantic completion (50% semantic labels). We train the models with semantic labels of video frames $\{I_1, I_2, I_3, I_{10}, I_{11}, I_{12}\}$ and predict novel semantic views on the rest frames $\{I_4, I_5, ..., I_9\}$.

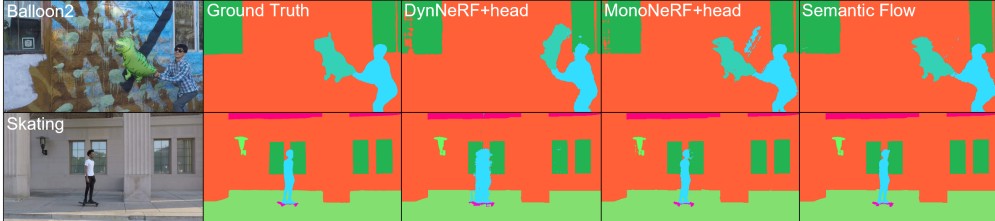

(d) Dynamic scene tracking (25% semantic labels). We use the semantic labels of frames $\{I_1, I_2, I_3\}$ to train the models, and conduct semantic predictions of novel views on the frames $\{I_4, I_5, ...I_{12}\}$.

Figure 3: Visualizations on various tasks. Different from DynNeRF (Gao et al., 2021)+semantic head and MonoNeRF (Tian et al., 2023)+semantic head that learn from point features, Semantic Flow learns from flow features for capturing motions. In this way, Semantic Flow predicts semantic labels of dynamic foregrounds with more accurate motions and clearer boundaries.

## 4  EXPERIMENTS

We evaluate our method by conducting experiments on various semantic tasks of dynamic scenes. We first introduce our new dataset and the implementation details. Then, we present experiments by training with full annotated labels. After that, we tested the performance on training with few labels.

### 4.1  DATASET AND IMPLEMENTATION DETAILS

**Dataset**. We introduce Semantic Dynamic Scene dataset which is built upon the Dynamic Scene dataset (Yoon et al., 2020). Dynamic Scene dataset contains 9 dynamic scenes captured by 12

Table 1: Performance on semantic representation learning from multiple scenes and semantic adaption of novel scenes (500 finetuning steps). $\diamond$ represents adding a semantic head to the original model. By learning from flow features, our model could reconstruct semantic fields and transfer to novel scenes with the state-of-the-art performance. More comparisons are in Table 9 in *Appendix*.

| Total Acc↑ / mIoU ↑ | Learning semantics from multiple scenes | | | | Semantic adaption of novel scenes | | |
|---|---|---|---|---|---|---|---|
| | Balloon1 | Balloon2 | Jumping | Skating | Umbrella | Playground | Truck |
| DynNeRF$^{\diamond}$ (Gao et al., 2021) | 0.767 / 0.515 | 0.459 / 0.229 | 0.912 / 0.570 | 0.955 / 0.431 | 0.941 / 0.795 | 0.914 / 0.387 | 0.967 / 0.662 |
| MonoNeRF$^{\diamond}$ (Tian et al., 2023) | 0.907 / 0.760 | 0.967 / 0.616 | 0.929 / 0.576 | **0.973** / 0.590 | 0.961 / 0.685 | 0.879 / 0.393 | 0.968 / 0.425 |
| Semantic Flow | **0.919 / 0.844** | **0.967 / 0.839** | **0.938 / 0.703** | 0.970 / **0.608** | **0.970 / 0.884** | **0.937 / 0.742** | **0.977 / 0.765** |

Table 2: Quantitative results on scene completion and dynamic scene tracking. $\diamond$ denotes that we added a semantic head to the original model. The results show the superiority of our model when learning from with few labels. Detailed results are shown in Table 10 and Table 11 in *Appendix*.

| method | Completion (50% labels) | | | Tracking (25% labels) | | |
|---|---|---|---|---|---|---|
| | Total Acc↑ | Avg Acc↑ | mIoU↑ | Total Acc↑ | Avg Acc↑ | mIoU↑ |
| DeAOT (Yang & Yang, 2022) | 0.816 | 0.600 | 0.412 | 0.776 | 0.533 | 0.359 |
| DynNeRF$^{\diamond}$ (Gao et al., 2021) | 0.934 | 0.849 | 0.738 | 0.896 | 0.760 | 0.660 |
| MonoNeRF$^{\diamond}$ (Tian et al., 2023) | 0.956 | 0.891 | 0.786 | 0.935 | 0.818 | 0.716 |
| Semantic Flow | **0.961** | **0.901** | **0.818** | **0.942** | **0.835** | **0.767** |

cameras with a camera rig. We manually annotate 7 dynamic scenes: Balloon1, Balloon2, Jumping, Skating, Playground, Truck and Umbrella. More details refer to Section D in *Appendix*.

**Implementation details**. We follow previous works (Yu et al., 2021; Liu et al., 2023a; Tian et al., 2023) to use MLPs with residual links as our flow network $\Psi_{flow}$ and geometry network $\Psi_{geo}$. The semantic networks $\Psi_{sem}^{dy}, \Psi_{sem}^{st}$ are implemented by three fully connected layers with ReLU activation. For the semantic field of dynamic foreground, we employ SlowOnly (Feichtenhofer et al., 2019) pretrained on Kinetics-400 dataset (Carreira & Zisserman, 2017) as our backbone encoder $E_{dy}$ with the frozen weights. We remove the final fully-connected layer and combined the first, second, and third feature layers as the output of $E_{dy}(I_t)$. We simplify Equation (6) where we only sample the flow feature vector $F(\Gamma)$ at timestamp $t$ from frames $\{I_{t-1}, I_t, I_{t+1}\}$. We train the semantic field of dynamic foreground for 40,000 iterations with Adam optimizer (Kingma & Ba, 2015). For the semantic field of static background, we employ ResNet18 (He et al., 2016) pretrained on ImageNet (Deng et al., 2009) as our backbone encoder. We pretrain the semantic field of static foreground for 100,000 iterations. The learning rate is set to $5 \times 10^{-4}$. More details refer to Section E in *Appendix*.

## 4.2 SEMANTIC VIEW SYNTHESIS WITH FULL LABELS

In this section, we evaluate our model with fully annotated labels. We first test the ability of semantic representation by learning from multiple scenes. Then, we conduct ablation studies and instance-level scene editing application based on the learned representation. Finally, we evaluate the generalization ability by finetuning our model on novel scenes.

**Learning semantics from multiple scenes.** Similar to Tian et al. (2023), we train our model on two scenes simultaneously: 1) Balloon1 and Balloon2; 2) Jumping and Skating. To compare with other state-of-the-art methods, we reimplement DynNeRF (Gao et al., 2021) and MonoNeRF (Tian et al., 2023) from their official implementation and add a semantic head to their models for predicting semantics. As shown in Figure 3a, because DynNeRF learns the semantic field from position embedding, it has limited generalization ability cross scenes, and hence the semantic information of two scenes is mixed together. As for MonoNeRF, due to the lack of motion information, the boundaries of dynamic objects are difficult to predict. In Table 1, since the mIoU matrix is sensitive to boundary accuracy, Semantic Flow outperforms the above two methods with a large margin.

**Ablation studies.** We conduct a series of ablation studies on Figure 4 and Table 6 by learning semantics from multiple scenes. It can be seen that with the proposed flow attention module, our model successfully extracts motion information in the flows and improves the performance on mIoU matrix. $\sigma_{dy}$ provides an important prior for integrating the semantic logits of dynamic objects. Our model renders the semantic field with finer details by using $L_{consist}$. $\Psi_{st}^{sem}$ and $\Delta\Gamma$ contribute to reconstructing the semantics in the foregrounds.

Table 3: Label percentages (Total Acc↑ / mIoU↑).

| Labels | Completion | Tracking |
|--------|-----------|----------|
| 25% | 0.979 / 0.904 | 0.975 / 0.893 |
| 50% | 0.983 / 0.920 | 0.979 / 0.908 |
| 75% | 0.985 / 0.934 | 0.985 / 0.934 |

Table 4: Flow displacements (Total Acc↑ / mIoU↑).

| Disp. | Completion | Tracking |
|-------|-----------|----------|
| $\Delta\Gamma$ | 0.984 / 0.925 | 0.975 / 0.893 |
| $\Gamma$ | 0.983 / 0.920 | 0.974 / 0.894 |
| $\Gamma \& \Delta\Gamma$ | 0.985 / 0.932 | 0.970 / 0.874 |

Table 5: Finetuning steps on novel scene. $\diamond$: semantic head.

| Ft steps | MonoNeRF$^\diamond$ | Semantic Flow |
|----------|---------------------|---------------|
| 100 | 0.819 / 0.296 | 0.906 / 0.689 |
| 200 | 0.869 / 0.316 | 0.925 / 0.736 |
| 500 | 0.879 / 0.393 | **0.937 / 0.742** |

Table 6: Numeric comparisons on $\sigma_{dy}$ prior, flow attention module, $\Psi_{st}^{sem}$, $L_{consist}$ and $\Delta\Gamma$.

| | Acc↑ / mIoU↑ |
|--------------------------|--------------|
| w/o. $\sigma_{dy}$ prior | 0.903 / 0.838 |
| w/o. flow attn | 0.921 / 0.643 |
| w/o. $\Psi_{st}^{sem}$ | 0.853 / 0.734 |
| w/o. $L_{consist}$ | 0.917 / 0.741 |
| w/o. $\Delta\Gamma$ | 0.915 / 0.770 |
| Semantic Flow | **0.932 / 0.859** |

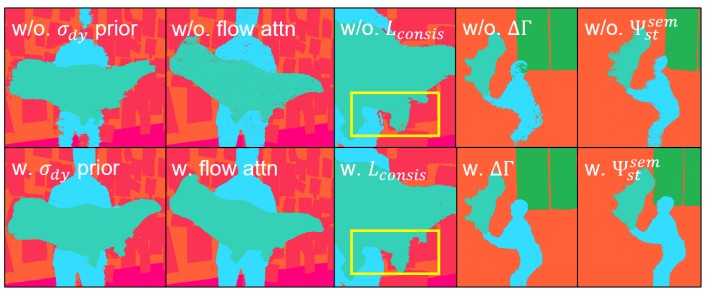

Figure 4: Ablation studies on $\sigma_{dy}$ prior, flow attention module, $L_{consist}$, $\Delta\Gamma$ and $\Psi_{st}^{sem}$.

**Instance-level scene editing.** Our model could conduct instance-level scene editing applications with the learned semantic field. As shown in Figure 1, our model can remove the dynamic instances in the foreground by forcing the volume densities of points related to target instances to zeros.

**Semantic adaption on novel scenes.** After training on multiple scenes, we finetune our model on unseen monocular videos to test the generalization ability on novel scenes. Specifically, we pre-train our model on Balloon1 and Balloon2 scenes with 10,000 training iterations and finetune the model on Umbrella, Playground and Truck scenes separately. We also evaluate the performance of DynNeRF (Gao et al., 2021) and MonoNeRF (Tian et al., 2023) with the same experiment settings for a fair comparison. As shown in Figure 3b, since our model learns semantics from flow features, it could predict more accurate boundaries of moving objects in dynamic foreground by capturing motion information. In contrast, the predictions from DynNeRF and MonoNeRF methods show blurring and even wrong boundaries, which causes many false positive predictions outside the moving objects and leads to a severe performance drop on mIoU matrix in Table 1 and Table 5.

### 4.3 SEMANTIC VIEW SYNTHESIS WITH FEW LABELS

Since manually annotating semantic labels is a time-consuming and laborious task, in this sections, we introduce two strategies to evaluate the generalization ability of our model with few labels: **(1) Semantic completion.** We apply models to learn the semantic representation over the entire scene with the semantic labels in the first and last few frames $\{I_1, I_2, I_3, I_{10}, I_{11}, I_{12}\}$. **(2) Dynamic scene tracking.** We train on the front video frames $\{I_1, I_2, I_3\}$ with fully annotated labels and directly infer the semantics over the rest frames. As shown in Figure 3c and Figure 3d, since DynNeRF builds the semantic field from position embedding, it cannot transfer to unseen frames with accurate semantic predictions. As MonoNeRF predicts the semantic labels with point features, it meets difficulties in predicting the semantic labels of the moving balloons in the foreground. In Table 2, with the extracted flow features, Semantic Flow is easier to identify the dynamic parts and hence presents higher performance. Table 3 and Table 4 show that our model could extract motions in flows from various types of displacements and build semantic fields by using 25% labels.

## 5 CONCLUSION

In this paper, we build a semantic field of dynamic scenes from monocular videos. We propose to learn from flow features that contain motion information and consider the volume densities as opacity prior for supervising the field with semantic labels on video frames. Experiments demonstrate the effectiveness of our model.

ACKNOWLEDGMENTS

This work was supported by the National Key Research and Development Program of China under Grant No. 2020AAA0108100, the National Natural Science Foundation of China under Grant Nos. 62206147, 62327808 and 62088102. The authors also would like to thank Dong Zhang, Zongwei Zhou, Wenxuan Li, Yaoyao Liu and Chuanruo Ning for their thoughtful suggestions.

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

# A  SUPPLEMENTAL VIDEO

We recommend readers to watch our supplemental video for more visual comparisons.

# B  CODES AND DATASET

The implementation codes of our model and our Semantic Dynamic Scene dataset are available at https://github.com/tianfr/Semantic-Flow/.

# C  LOSS

Similiar to MonoNeRF (Tian et al., 2023), we use RGB loss and optical flow loss to supervise the static and dynamic radiance fields. Let $L_{st}^{rgb}, L_{dy}^{rgb}$ denote the RGB color loss of static and dynamic field separately, $L_{full}^{rgb}$ denote the RGB color loss when jointly optimizing the static and dynamic radiance fields, and $L_{opt}$ denote the optical flow loss. The entire loss of the model is defined as

$$L = \alpha_{st}^{rgb} L_{st}^{rgb} + \alpha_{dy}^{rgb} L_{dy}^{rgb} + \alpha_{full}^{rgb} L_{full}^{rgb} + \alpha_{opt} L_{opt} + \alpha_{full}^{sem} L_{full}^{sem} + \alpha_{dy}^{sem} L_{dy}^{sem}$$
$$+ \alpha_{st}^{sem} L_{st}^{sem} + \alpha_{consist} L_{consist}, \tag{17}$$

where the hyper-parameters of each loss is listed in Table 7.

| Param | $\alpha_{st}^{rgb}$ | $\alpha_{dy}^{rgb}$ | $\alpha_{full}^{rgb}$ | $\alpha_{opt}$ | $\alpha_{full}^{sem}$ | $\alpha_{dy}^{sem}$ | $\alpha_{st}^{sem}$ | $\alpha_{consist}$ |
|---|---|---|---|---|---|---|---|---|
| Value | 4 | 1 | 1 | 0.02 | 0.16 | 0.08 | 0.08 | 0.01 |

Table 7: Hyper-parameters of loss.

# D  ANNOTATION DETAILS

We annotate 7 dynamic scenes from Dynamic Scene dataset (Yoon et al., 2020). For each scene, we classify two different types of semantic labels: foreground labels and background labels. All the foreground semantic labels in one scene formulate the foreground mask of the scene. The label details of each scene is shown in Table 8. The examples of annotated images are shown in Figure 5. We follow previous works (Tian et al., 2023; Gao et al., 2021) and use video frames from different cameras to simulate the camera movement. All the cameras capture images at 12 different timestamps. During the training, each monocular video contains 12 frames, where the $t$-th frame is sampled from $t$-th camera at time $t$.

Table 8: Scene labels in our dataset.

| Scene | Foreground labels | Background labels |
|---|---|---|
| Jumping | person1, person2, person3, person4, | bag, lamp, ground, window, red_wall, gray_wall |
| Skating | person, skate | lamp, ground, window, red_wall, gray_wall |
| Truck | truck | car, ground, building |
| Umbrella | umbrella | red_wall, plants, ground |
| Balloon1 | person, balloon | red_wall grey_wall, newspaper |
| Balloon2 | person, balloon | red_wall grey_wall, window |
| Playground | person, balloon | gym_device, ground |

# E  IMPLEMENTATION DETAILS

In this section, we introduce the details of our Semantic Flow and experimental settings. The entire model is trained on a Nvidia RTX 3090 GPU with a total batch size of 1024 rays. The learning rate is 0.0005. We used Adam optimizer (Kingma & Ba, 2015) where betas is $(0.9, 0.999)$. During the training, we first pretrain the semantic field of static background, and optimize the semantic field of dynamic foreground with the pretrained static semantic field.

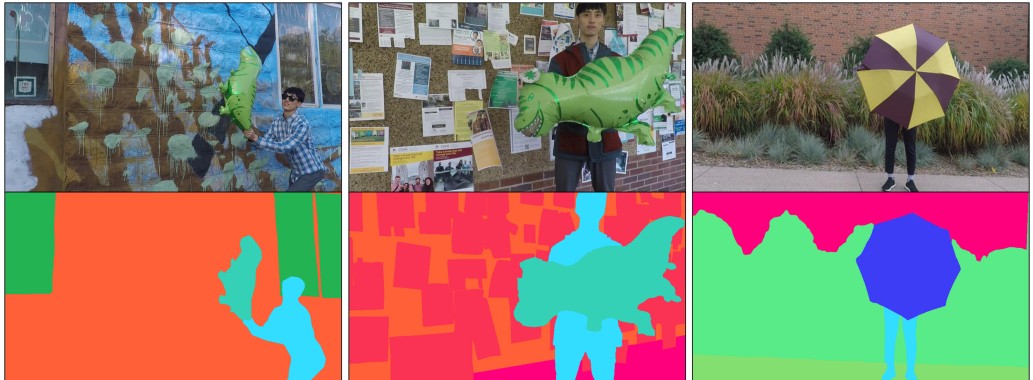

Figure 5: Annotation examples in Semantic Dynamic Scene dataset.

**Details of semantic fields for dynamic foreground**. We use the SlowOnly50 network (Feichtenhofer et al., 2019) as our encoder $E_{dy}$ and remove all the pooling layers in the network. The encoder is pretrained on Kinetics-400 dataset (Carreira & Zisserman, 2017). We remove the final fully-connected layer in the encoder. We froze the pretrained weights and the point feature vector $\boldsymbol{F}_{dy}(I; \Phi_t)$ is sampled from the first, second and third spatial feature maps of the encoder, which are concatenated in the channel dimension and fused with a fully connected layer. The vector has 256 channels. We follow previous works (Gao et al., 2021; Tian et al., 2023) and use the same embedding function $\gamma(\cdot)$. We follow Tian et al. (2023) and implement the flow network $\Psi_{flow}$ and geometric network $\Psi_{geo}$ by using MLPs with residual links and width 128. We employ 4 residual blocks to implement $\Psi_{flow}$ and $\Psi_{geo}$. The semantic network $\Psi_{sem}^{dy}$ is implemented by three fully connected layers with ReLU activation. As for flow attention module, the head number $H$ is 4 and the number of channels $C$ is 64. The inital training steps is 40,000 for each scene.

**Details of semantic field for static backgrounds**. We follow Tian et al. (2023) and use ResNet18 (He et al., 2016) as our encoder $E_{st}$. The point feature $F_{st}(I; \boldsymbol{x})$ is sampled from four feature maps prior to four pooling layers of ResNet18 and concatenated in the channel dimension. We fused the concatenated features with a fully connected layer to form a feature vector of size 256. The semantic network $\Psi_{st}^{sem}$ is implemented by three fully connected layers with ReLU activation. The initial training steps is 100,000 for each scene.

**Details of learning from multiple scenes**. We train our semantic field on two scenes simultaneously. During training, there are 512 rays for each scene in a mini-batch. It takes about 16 hours to learn the semantic field from two scenes.

**Details of semantic adaption on novel scenes**. After training from multiple scenes, we finetune our model on novel scenes by using the pretrained model on Balloon1 and Balloon2 scenes with 10,000 training steps. The semantic fields of foregrounds and backgrounds are separately finetuned with preferred finetuning steps. It takes about 10 minutes to finetune the model on a novel scene with 500 steps.

**Scene editing**. We conduct instance-level scene editing applications by control the volume densities related to specific semantic labels. For instance, we delete the *person1* in the Jumping scene by forcing the volume densities of the points that has the label of *person1* to zeros.

**Details of semantic completion**. In this setting, after training the model on the frames that include semantic labels, we directly employ the symantic view synthesis on the rest unseen frames. We change the initial flow displacement to $\boldsymbol{\Gamma} \& \Delta\boldsymbol{\Gamma}$.

**Details of dynamic scene tracking**. Similar to semantic completion, we directly conduct semantic view synthesis on the unseen frames by using the pretrained model on the frames with semantic labels. The initial flow displacement here is $\Delta\boldsymbol{\Gamma}$.

Table 9: More comparisons on semantic representation learning from multiple scenes and semantic adaption of novel scenes (500 finetuning steps). $\diamond$ represents adding a semantic head to the original model. By learning from flow features, our model could reconstruct semantic fields and transfer to novel scenes with the state-of-the-art performance.

| Total Acc↑ / mIoU ↑ | Learning semantics from multiple scenes | | | | Semantic adaption of novel scenes | | |
|---|---|---|---|---|---|---|---|
| | Balloon1 | Balloon2 | Jumping | Skating | Umbrella | Playground | Truck |
| NeRF+time$^\diamond$ (Mildenhall et al., 2020) | 0.780 / 0.529 | 0.455 / 0.211 | 0.913 / 0.564 | 0.857 / 0.277 | 0.953 / 0.763 | 0.919 / 0.416 | 0.920 / 0.433 |
| NSFF$^\diamond$ (Li et al., 2021) | 0.583 / 0.402 | 0.511 / 0.224 | 0.865 / 0.420 | 0.891 / 0.337 | 0.954 / 0.773 | 0.910 / 0.623 | 0.918 / 0.614 |
| RoDyNeRF$^\diamond$ (Liu et al., 2023b) | 0.567 / 0.459 | 0.531 / 0.312 | 0.886 / 0.432 | 0.888 / 0.536 | 0.900 / 0.312 | 0.876 / 0.432 | 0.904 / 0.606 |
| D-NeRF$^\diamond$ (Pumarola et al., 2020) | 0.267 / 0.079 | 0.381 / 0.076 | 0.267 / 0.135 | 0.305 / 0.136 | 0.700 / 0.221 | 0.605 / 0.323 | 0.780 / 0.501 |
| DynNeRF$^\diamond$ (Gao et al., 2021) | 0.767 / 0.515 | 0.459 / 0.229 | 0.912 / 0.570 | 0.955 / 0.431 | 0.941 / 0.795 | 0.914 / 0.387 | 0.967 / 0.662 |
| MonoNeRF$^\diamond$ (Tian et al., 2023) | 0.907 / 0.760 | 0.967 / 0.616 | 0.929 / 0.576 | **0.973** / 0.590 | 0.961 / 0.685 | 0.879 / 0.393 | 0.968 / 0.425 |
| Semantic Flow | **0.919 / 0.844** | **0.967 / 0.839** | **0.938 / 0.703** | 0.970 / **0.608** | **0.970 / 0.884** | **0.937 / 0.742** | **0.977 / 0.765** |

# F    DETAILED RESULTS ON SEMANTIC COMPLETION AND DYNAMIC SCENE TRACKING

Table 10: Quantitative results on semantic completion. We train the semantic field with 50% frames with semantic labels and tested the semantic view synthesis performance on the rest frames. $\diamond$ denotes that we added a semantic head to the original model.

| Acc↑ / mIoU ↑ | Jumping | Skating | Truck | Umbrella | Balloon1 | Balloon2 | Playground | Average |
|---|---|---|---|---|---|---|---|---|
| DeAOT (Yang & Yang, 2022) | 0.773 / 0.292 | 0.863 / 0.344 | 0.932 / 0.566 | 0.877 / 0.520 | 0.567 / 0.324 | 0.837 / 0.424 | 0.866 / 0.415 | 0.816 / 0.412 |
| DynNeRF$^\diamond$ (Gao et al., 2021) | 0.896 / 0.660 | 0.955 / 0.660 | 0.949 / 0.719 | 0.953 / 0.820 | 0.883 / 0.782 | 0.947 / 0.784 | 0.956 / 0.745 | 0.934 / 0.738 |
| MonoNeRF$^\diamond$ (Tian et al., 2023) | 0.937 / 0.725 | 0.964 / 0.701 | 0.975 / 0.769 | 0.966 / 0.844 | 0.912 / 0.820 | 0.978 / 0.899 | **0.962** / 0.746 | 0.956 / 0.786 |
| SemanticFlow | **0.940 / 0.771** | **0.981 / 0.777** | **0.976 / 0.786** | **0.967 / 0.860** | **0.922 / 0.843** | **0.983 / 0.920** | 0.961 / **0.769** | **0.961 / 0.818** |

Table 11: Quantitative results on dynamic scene tracking. We train the model on the front 25% frames with semantic labels and evaluated the performance by render novel semantic views on the rest frames. $\diamond$ represents that we add a semantic head to the original model.

| Acc↑ / mIoU ↑ | Jumping | Skating | Truck | Umbrella | Balloon1 | Balloon2 | Playground | Average |
|---|---|---|---|---|---|---|---|---|
| DeAOT (Yang & Yang, 2022) | 0.695 / 0.200 | 0.784 / 0.222 | 0.915 / 0.524 | 0.797 / 0.412 | 0.560 / 0.325 | 0.824 / 0.393 | 0.855 / 0.437 | 0.776 / 0.359 |
| DynNeRF$^\diamond$ (Gao et al., 2021) | 0.856 / 0.593 | 0.912 / 0.563 | 0.914 / 0.637 | 0.926 / 0.767 | 0.823 / 0.689 | 0.925 / 0.729 | 0.922 / 0.648 | 0.896 / 0.660 |
| MonoNeRF$^\diamond$ (Tian et al., 2023) | 0.896 / 0.619 | 0.964 / 0.624 | 0.928 / 0.623 | 0.954 / 0.762 | 0.897 / 0.798 | 0.954 / 0.791 | **0.954 / 0.797** | 0.935 / 0.716 |
| Semantic Flow | **0.906 / 0.676** | **0.966 / 0.665** | **0.946 / 0.688** | **0.955 / 0.866** | **0.902 / 0.806** | **0.975 / 0.893** | 0.950 / 0.772 | **0.942 / 0.767** |

**Semantic completion**. We list the detailed performance of semantic completion in Table 10. The results are reported by training with 50% labels.

**Dynamic Scene tracking**. We present the detailed results of dynamic scene tracking in Table 11. The experiments are conducted by training the model with semantic labels on frames $\{I_1, I_2, I_3\}$ *i.e.,* 25% labels.

# G    DISCUSSIONS

## G.1    DISCUSSION ABOUT SCENE RECONSTRUCTION

In this section, we compare the scene reconstruction performance of our model with MonoNeRF (Tian et al., 2023) and DynNeRF (Gao et al., 2021). Table 12 shows that our model achieves better performance in PSNR, SSIM and LPIPS indexes. By reconstructing the dynamic scenes precisely, our model could conduct instance-level scene editing applications.

## G.2    DISCUSSION ABOUT FLOW FIELD AND CORRESPONDENCE

We visualize the predicted flow fields and pixel correspondence of the dynamic foreground in three consecutive frames in Figure 6. Although we do not add extra restrictions to the mappings of flows in (16), the visualization of the flow fields and correspondence shows that our model is able to predict flows that map similar parts of the dynamic foreground across time.

Table 12: The results of scene reconstruction. Our model achieves better color rendering performance compared with other state-of-the-art methods.

| PSNR ↑ / SSIM ↑ / LPIPS ↓ | Jumping | Skating | Average |
|---|---|---|---|
| DynNeRF (Gao et al., 2021) | 21.91 / 0.6856 / 0.174 | 24.68 / 0.7866 / 0.175 | 23.30 / 0.7361 / 0.176 |
| MonoNeRF (Tian et al., 2023) | 22.41 / 0.7484 / 0.145 | **26.18** / 0.8739 / 0.115 | 24.30 / 0.8112 / 0.130 |
| Semantic Flow | **22.86 / 0.7658 / 0.138** | 25.75 / **0.8774 / 0.113** | **24.31 / 0.8216 / 0.126** |

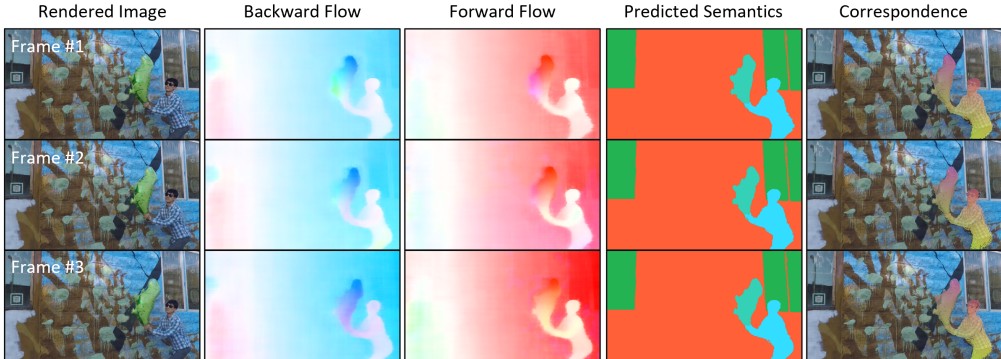

Figure 6: Visualization of rendered RGB images, estimated flow fields, semantic predictions and pixel correspondence in three consecutive frames. In the correspondence visualization, the color coding illustrates correspondences of the dynamic foreground across time.

### G.3 DISCUSSION ABOUT OPTICAL FLOWS

Table 13: Performance on semantic representation learning from multiple scenes with optical flow maps generated from different flow estimation methods (Total Acc↑ / mIoU↑).

| method | Learning semantics from multiple scenes | | | |
|---|---|---|---|---|
| | Balloon1 | Balloon2 | Jumping | Skating |
| RAFT (Teed & Deng, 2020) | 0.919 / 0.844 | 0.967 / 0.839 | 0.936 / 0.733 | 0.970 / 0.716 |
| FlowNet (Dosovitskiy et al., 2015) | 0.926 / 0.855 | 0.974 / 0.766 | 0.921 / 0.636 | 0.965 / 0.591 |

Table 14: Performance on semantic representation learning from multiple scenes under the supervision of noisy optical flow maps (Total Acc↑ / mIoU↑).

| noise scale $\beta$ | Learning semantics from multiple scenes | | | |
|---|---|---|---|---|
| | Balloon1 | Balloon2 | Jumping | Skating |
| 0 | 0.919 / 0.844 | 0.967 / 0.839 | 0.936 / 0.733 | 0.970 / 0.716 |
| 1% | 0.923 / 0.844 | 0.963 / 0.820 | 0.937 / 0.715 | 0.970 / 0.707 |
| 5% | 0.924 / 0.808 | 0.964 / 0.813 | 0.937 / 0.656 | 0.969 / 0.714 |
| 10 % | 0.922 / 0.777 | 0.964 / 0.666 | 0.936 / 0.690 | 0.969 / 0.670 |

Since Semantic Flow relies on the optical flows predicted from the pretrained models, in this section, we discuss the robustness of the model supervised by optical flow signals from two perspectives: choosing different optical flow estimation methods and adding noise to optical flow maps. We first compare the semantic prediction performance by using RAFT (Teed & Deng, 2020) or FlowNet (Dosovitskiy et al., 2015). accroding to the experiments in RAFT paper, RAFT outperforms FlowNet with a large margin in various optical flow estimation tasks. Therefore, it can be seen that there is about 10% performance drop in mIOU matrix when jointly optimizing Jumping and Skating scenes, where the dynamic foregrounds are drastically changing. However, our model still reaches comparable results when jointly optimizing Balloon1 and Balloon2 scenes with the flow maps estimated from FlowNet. We also test the performance of our model by manually adding different scales of noise to the predicted optical flow maps. Concretely, for each optical flow map estimated from two consecutive video frames, we denote $\Phi_{min}, \Phi_{max}$ as the minimum and maximum numbers of the optical flows in the map, and the noisy flow can be defined as the following

equation,

$$\mathbf{\Phi}_{noisy} = \mathbf{\Phi}_{origin} + \beta \times \boldsymbol{x}_{noise}, \tag{18}$$

where $\beta$ controls the scale of the noise and $\boldsymbol{x}_{noise} \sim \mathcal{U}(\mathbf{\Phi}_{min}, \mathbf{\Phi}_{max})$ denotes the noise. Table 14 presents that our model shows robustness with a small scale of noise on the optical flow maps.

### G.4 DISCUSSION ABOUT OCCLUSION

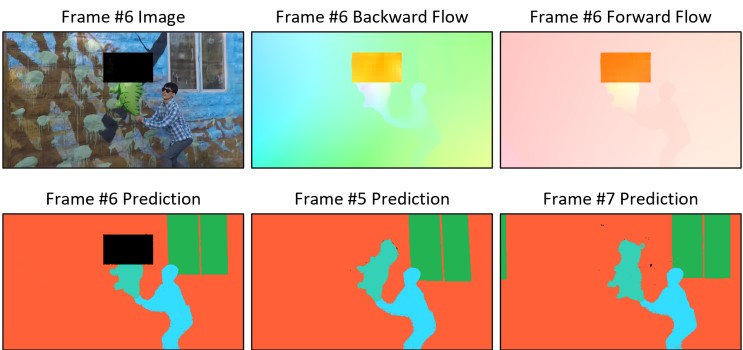

Figure 7: Visualization of the occlusion situation. We manually occlude a part of both static and dynamic regions in frame #6, which leads to the wrong flow maps estimated by Teed & Deng (2020). Although Semantic Flow fails to predict the semantics of the occluded part in frame #6, it successfully conducts accurate predictions in frames #5 and #7.

To test the performance of our model with occluded regions, we manually add an occluded region to the frame #6 in the Balloon2 scenes. Concretely, as shown in Figure 7, we manually occlude a region in the RGB image and semantic labels of the frame #6 and generate the wrong flow maps from the occluded image by using Teed & Deng (2020). We train the model with the occluded image and wrong optical flow maps. Although Semantic Flow meets difficulty in predicting the semantics of the occluded region in the frame #6, it successfully predicts the semantics in the frames #5 and #7.

### G.5 COMPARED TO IMAGE SEGMENTATION METHOD

To compare with the image segmentation method, we use the pretrained Masked R-CNN (He et al., 2017) model to predict the semantic labels in the Jumping scene. As shown in Figure 8, while Masked R-CNN processes each video frame independently and hence predicts inconsistent labels of the same instances in different timestamps, our model could generate semantic labels consistent in time by learning semantics from all the video frames.

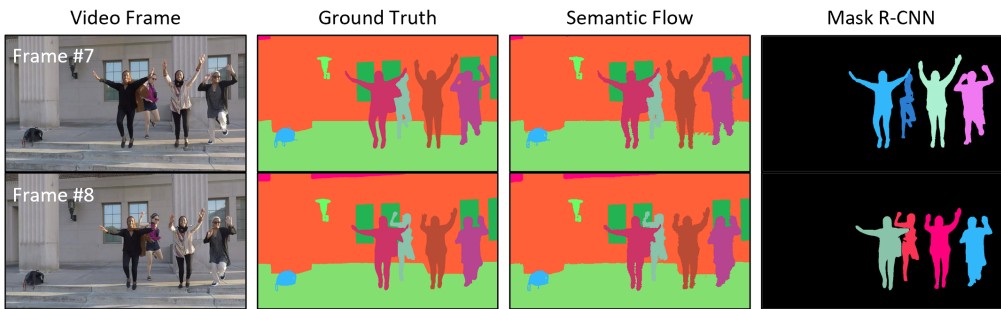

Figure 8: Comparison with Masked-RCNN (He et al., 2017). Since Semantic Flow learns a semantic field that is continuous in time, it predicts consistent semantic labels at all timestamps. In contrast, image segmentation methods such as Masked-RCNN (He et al., 2017) process each video frame individually, and hence align inconsistent instance-level labels to the same instance in different frames.

### G.6 Discussion about Flow Displacements

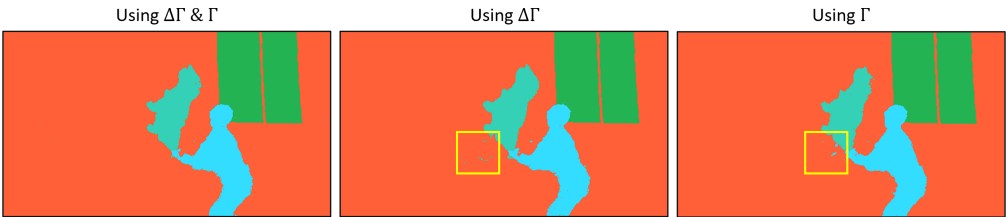

Figure 9: Qualitative results by using different displacements in the dynamic scene tracking setting. While Table 4 shows that our model has robustness with different displacements, we find that our model could predict semantics with better visualization quality by using $\Gamma \& \Delta\Gamma$.

In Table 6 and Figure 4 in the main paper, we demonstrate that the boundaries of predicted semantics could be more precise and the performance could be improved by using flow displacements. While in Table 4 we show that our model has robustness with different flow displacements, we find that it could generate better qualitative results by using $\Gamma \& \Delta\Gamma$, as shown in Figure 9.

### G.7 Limitations

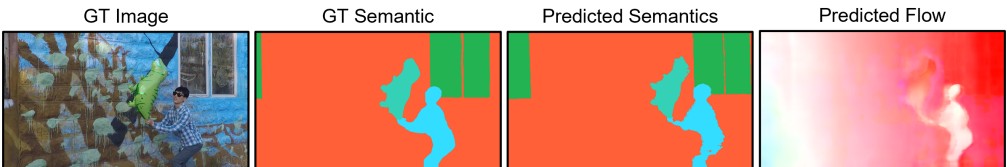

Figure 10: A failure case in the tracking setting. Our model fails to predict precise semantics with imperfect flow predictions.

It is worth noting that training semantic fields with few semantics labels is still a challenging problem. For instance, in the dynamic scene tracking setting, Semantic Flow may fail to conduct precise semantic prediction due to imperfect flow predictions as shown in Figure 10.

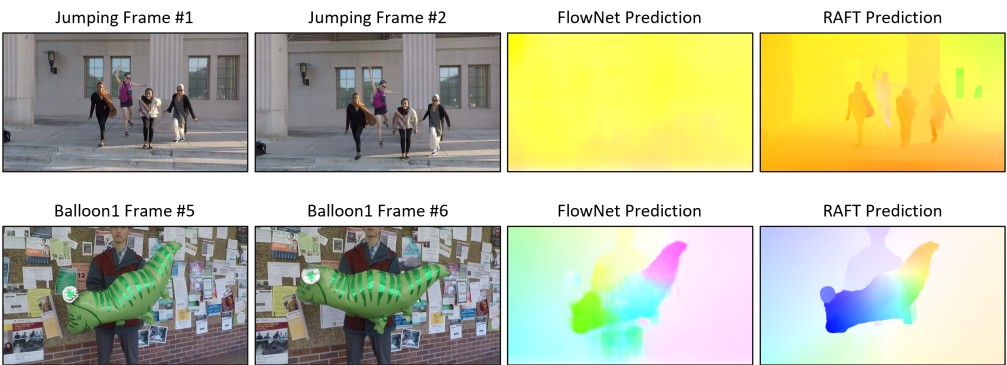

Figure 11: Comparison of the flow prediction between FlowNet (Dosovitskiy et al., 2015) and RAFT (Teed & Deng, 2020). In the Jumping scene, the flow maps predicted by FlowNet are totally wrong and lead to a performance drop in Table 13. On the other hand, in the Balloon1 scene, while the flow maps predicted by FlowNet are less accurate than RAFT, it successfully predicts the boundary of the balloon movement. The inaccuracy in the maps can be considered as a small scale of noise, which may slightly improve the performance in the Balloon1 scene as shown in Table 14.

