# OpenReview forum: "Semantic Flow: Learning Semantic Fields of Dynamic Scenes from Monocular Videos"
_ICLR.cc/2024/Conference — ICLR 2024 poster_

### Official Review · Reviewer_QtRh · 2023-10-31

**Soundness:** 3 good
**Presentation:** 3 good
**Contribution:** 2 fair
**Rating:** 5
**Confidence:** 4

**Summary:**

In this paper the problem of novel view synthesis of semantic labels is studied. Rather than rendering colour, the segmentation is rendered with a NeRF model. The authors propose a model and training procedure that can learn scene flow fields and semantic renderings of a given video sequence. Furthermore, the proposed method allows for quick adaptation on novel video sequences. Since no earlier work studies this problem, the authors label the nvidia dynamic scenes dataset with pixel-wise semantic labels. The experiments show that the semantic labels can be rendered accurately, and furthermore that we do not need labels for all frames in a sequence, and that we can use the semantic labels to mask out specific parts of the video and render it without specific objects.

**Strengths:**

- There is no existing dataset for the problem setup, so authors annotated the nvidia dynamic scenes dataset with pixel-wise semantic segmentation labels.
- A strength of the method is that it allows for quick adaptation to new scenes, e.g. with just 500 iterations it can perform well given pre-training on other scenes. The reason is that the scene flow field is not learned from scratch, but rather from frame-wise video features, which does not need to be learned from scratch for each scene.
- Augmenting NeRF models with semantic segmentation has been done for static scenes (e.g. Zhi et al 2021) but to the best of my knowledge not for general dynamic scenes, so the paper tackles a new problem setup.
- The method is clearly described and ablations are provided for the main components.

**Weaknesses:**

- There are some baselines that would be reasonable to try that are missing from the paper. For instance, if we just render the rgb images with any NeRF method (e.g. MonoNeRF) and apply some video object segmentation algorithm (e.g. any top-performing method on the DAVIS dataset) or semantic segmentation method (trained on some dataset with overlapping labels), how well would that perform?
- Since one of the applications mentioned in the paper is scene editing, i.e. removing some specific object, it is necessary to not just render semantics correctly but also rgb. There are no values provided for the standard novel view synthesis metrics (PSNR, SSIM, LPIPS) for rgb on the tested video sequences.
- It would have been interesting to somehow visualise or discuss the flow fields. Since the objective is semantic rendering rather than colour rendering it is not clear if we need the scene flow to map to the same specific part of an object, or if it is sufficient or even beneficial to just map to anywhere within the same object. For instance, the consistency loss L_consis only enforces that points along flow trajectories should have the same label.

Minor issues:
- Missing related work: “Panoptic neural fields: A semantic object-aware neural scene representation” (CVPR 2022) also considers novel view synthesis for semantic segmentation from a video, although their method is limited to non-deformable dynamic objects.
- Page 5: Does ground truth flow mean optical flow estimated from RAFT? If that is the case it should not be called ground truth.
- Page 5: The closing parenthesis in eq. (7) is probably incorrect.
- Page 8: “Boundray” typo
- Page 8: Table 2 caption: “qualitative” should be “quantitative”

**Questions:**

See everything under weaknesses.
- When training for semantic completion or tracking, only a subset of the frames are used for semantic supervision. Is the same true for RGB supervision or are all frames used for that?
- In Fig. 3, what are the indices of the frames that are shown? How far are they from the frames with semantic labels?
- For the DynNeRF and MonoNeRF baselines, what exactly is the input to the semantic heads that are learned?

---

> ### Author Response · Authors · 2023-11-19
> **Response to Reviewer QtRh**
>
> # Missing baselines
> Thank you for your question. During the rebuttal phrase, we test the semantic segmentation performance by applying the state-of-the-art tracking method DeAOT [1] and image segmentation method Mask R-CNN [2] to the rendered images from MonoNeRF [3]. The results are shown **in Table 2 and Figure 8**. We also discuss the performance of these methods in the **General Response**. DeAOT lacks semantic understanding of 3D scenes and hence predicts inconsistent semantic labels in different views, leading to a severe performance drop as shown in Table 2. As Mask R-CNN only learns semantic information from single images, Figure 8 shows that it predicts inconsistent semantic labels of the same instances in different frames.
>
> In addition, we also compare more Dynamic NeRF baselines by manually adding semantic segmentation heads to these methods in the **General Response and Table 1 in the paper**.
>
> # RGB rendering preformance
> Thank you for your question. We present the PSNR, SSIM and LPIPS scores of rendered RGB images **in the following table**. Our model could achieve slightly better RGB color rendering quality compared to MonoNeRF [3] and DynNeRF [4], which demonstrates that our model is able to conduct the instance-level scene editing application. We also add the results to **Section G.1 and Table 11 in the Appendix**.
>
> |PSNR / SSIM / LPIPS|  Jumping   | Skating  | Average |
> |----|----|----|----|
> |DynNeRF [4] + head|  21.91 / 0.6856 / 0.174    | 24.68 / 0.7866 / 0.175  |23.30 / 0.7361 / 0.176|
> |MonoNeRF [3] + head| 22.41 / 0.7484 / 0.145  | **26.18** / 0.8739 / 0.115 |24.30 / 0.8112 / 0.130|
> |Semantic Flow | **22.86** / **0.7658** / **0.138**  | 25.75 / **0.8774** / **0.113** | **24.31** / **0.8216** / **0.126**|
>
> # Visualization and discussion of the flow fields
> **In Section G.2 and Figure 6 in the Appendix**, we show the visualization of the rendered images, estimated flow fields, predicted semantics and the point correspondence of dynamical foreground in 3 consecutive frames. We also discuss these results **in the General Response**. We agree that we do not force the flows to map the same parts of an object moving in time. However, Figure 6 shows that our model is able to predict flows that map similar parts of the dynamic foreground.
>
> # Minor issues
> * *Missing related work*. We have inserted the Panoptic Neural Field [5] into **the Related Work section in the paper, page 3**.
> * *Ground truth flow*. Thank you for your suggestion. we have corrected "the ground truth flows" to "the flows generated from the pretrained model".
> * *Incorrect parenthesis*. Thanks. We have revised the closing parenthesis of $\boldsymbol{F}_{dy}(I;{\boldsymbol{\Gamma}})$ near the equation (7) in the paper.
> * *"Boundray" typo.* Thank you. We have revised "boundray" to "boundary" in page 8.
> * *Table 2 caption*. Thanks. We have revised "qualitative" to "quantitative" in Table 2 caption.
>
> # Questions
> * *RGB frames in semantic completion and tracking settings.* We follow MonoNeRF [3] and use the same subset of frames for RGB supervision.
> * *Indices of the frames in Figure 3*. In the completion setting, Figure 3 shows the semantic prediction results from the frame #8, which are 2-frame far from the frame #10 with the semantic labels. For the dynamic scene tracking setting, Figure 3 shows the results from the frame #10, which are 7-frame far from the frame #3 with the semantic labels.
> * *The input to the semantic heads.* For the DynNeRF [4] and MonoNeRF [3] methods, the input is the feature vector with 256 channels obtained from a 8-layer MLP with ReLU activations.
>
> References:
>
> [1] Yang, Zongxin, and Yi Yang. "Decoupling features in hierarchical propagation for video object segmentation." In NIPS, pp. 36324-36336, 2022.
>
> [2] He, Kaiming, Georgia Gkioxari, Piotr Dollár, and Ross Girshick. "Mask R-CNN." In ICCV, pp. 2961-2969. 2017.
>
> [3] Tian, Fengrui, Shaoyi Du, and Yueqi Duan. "Mononerf: Learning a generalizable dynamic radiance field from monocular videos." In ICCV, 2023.
>
> [4] Gao, Chen, Ayush Saraf, Johannes Kopf, and Jia-Bin Huang. "Dynamic view synthesis from dynamic monocular video." In ICCV, pp. 5712-5721. 2021.
>
> [5] Kundu, Abhijit, Kyle Genova, Xiaoqi Yin, Alireza Fathi, Caroline Pantofaru, Leonidas J. Guibas, Andrea Tagliasacchi, Frank Dellaert, and Thomas Funkhouser. "Panoptic neural fields: A semantic object-aware neural scene representation." In CVPR, pp. 12871-12881. 2022.

---

> ### Comment · Reviewer_QtRh · 2023-11-22
>
> Thank you for the answers. The additional baseline experiments strengthen the paper, although for Mask R-CNN it would be easy to use e.g. the Hungarian algorithm to match instances in consecutive frames as is common practice for video instance segmentation. The RGB rendering performance seems in line with the baselines, although it is only reported for two sequences in the Nvidia dynamic scenes datasets so it can not be easily compared to other methods than the reported ones. The flow fields look as expected, since optical flow was used for supervision.

---

> ### Author Response · Authors · 2023-11-22
>
> Thank you for your response!
>
> The superiority of our model compared to image-based or video-based segmentation methods is that our model predicts view-agnostic semantic labels of each point in the 3D scenes, while image-based or video-based segmentation methods lack the semantic understanding of the 3D scenes and hence predict inconsistent semantic labels across different views, which may lead to a severe performance drop. We still agree that we should evaluate the performance of Mask R-CNN + Hungarian for a much fairer comparison. **Currently we are testing the performance of Mask R-CNN + Hungarian algorithm and we will report the results as soon as possible.**
>
> The reason why we report the RGB rendering performance on the joint learning of Jumping and Skating scenes is that the examples of instance-level editing in the paper are conducted in this setting. We totally agree that we should report the performance of the entire dataset. **In the following table**, we report the RGB rendering performance on the entire Nvidia Dynamic Scene dataset. We follow the experimental setting of learning from multiple scenes in the MonoNeRF paper. It can be seen that our model reaches slightly better performance compared to MonoNeRF.
> |method|  PSNR   | SSIM  | LPIPS |
> |----|----|----|----|
> |DynNeRF [4] + head|  21.19    | 0.6226  | 0.304|
> |MonoNeRF [3] + head| 23.10  | 0.7441 | 0.170|
> |Semantic Flow | **23.47**  | **0.7524** | **0.165** |
>
> We also report all the details of the PSNR, SSIM and LPIPS score **in the following tables**.
>
> | PSNR          | Balloon1 | Balloon2 | Jumping | Skating | Umbrella | Playground | Truck | Average     |
> |---------------|----------|----------|---------|---------|----------|------------|-------|-------------|
> | DynNeRF [4] + head  | 19.17    | 18.98    | 21.91   | 24.68   | 20.67    | 20.57      | 22.38 | 21.19 |
> |MonoNeRF [3] + head  | 21.19    | 24.24    | 22.41   | 26.18   | 21.69    | 20.32      | 25.66 | 23.10 |
> | semantic flow | 21.15    | 24.12    | 22.86   | 25.75   | 22.26    | 21.86      | 26.32 | 23.47 |
>
>
> | SSIM          | Balloon1 | Balloon2 | Jumping | Skating | Umbrella | Playground | Truck  | Average     |
> |---------------|----------|----------|---------|---------|----------|------------|--------|-------------|
> | DynNeRF [4] + head| 0.5493   | 0.4690    | 0.6856  | 0.7866  | 0.5997   | 0.6435     | 0.6246 | 0.6226|
> |MonoNeRF [3] + head| 0.6733   | 0.7210    | 0.7484  | 0.8739  | 0.6672   | 0.7450      | 0.7800   | 0.7441 |
> | Semantic Flow | 0.6613   | 0.7053   | 0.7658  | 0.8774  | 0.6805   | 0.7614     | 0.8151 | 0.7524      |
>
> | LPIPS         | Balloon1 | Balloon2 | Jumping | Skating | Umbrella | Playground | Truck | Average     |
> |---------------|----------|----------|---------|---------|----------|------------|-------|-------------|
> |DynNeRF [4] + head| 0.381    | 0.486    | 0.174   | 0.175   | 0.332    | 0.291      | 0.290  | 0.304 |
> | MonoNeRF [3] + head | 0.204    | 0.176    | 0.145   | 0.115   | 0.201    | 0.183      | 0.167 | 0.170 |
> | Semantic Flow | 0.218    | 0.191    | 0.138   | 0.113   | 0.195    | 0.147      | 0.152 | 0.165 |

---

> ### Author Response · Authors · 2023-11-23
>
> Thank you for waiting!
>
> To test the performance of the Mask R-CNN + Hungarian algorithm, we choose **person, truck and umbrella** classes in our dataset, which overlap with the COCO dataset. We use the Mask R-CNN pretrained on the COCO dataset to predict semantic labels in each video frame and use the Hungarian algorithm to match the same instances in different frames. We present the results **in the following table**. Our model outperforms Mask R-CNN + Hungarian algorithm by a large margin, which demonstrates the superiority of our model that leans view-agnostic semantic labels in 3D space.
>
> |method|  Total Acc   | Avg Acc  | mIOU |
> |----|----|----|----|
> |Mask R-CNN + Hungarian|  0.689   | 0.482  | 0.308|
> |Semantic Flow | **0.964**  | **0.725** | **0.623** |

---

> > ### Author Response · Authors · 2023-11-23
> >
> > Thanks again for your valuable feedback and suggestions. We are wondering if the above responses have addressed your concerns. If there is any information or clarification you need from us, please feel free to let us know. We are more than happy to have further discussions!

---

### Official Review · Reviewer_5VQj · 2023-10-31

**Soundness:** 3 good
**Presentation:** 3 good
**Contribution:** 3 good
**Rating:** 8
**Confidence:** 3

**Summary:**

The paper looks to solve the problem related to generating a novel view 2D semantic map, for dynamic scenes using continuous flow. Paper leverages optical flow for the foreground part of the images and uses volume density as a prior to determining flow feature contribution towards semantics. Authors evaluate this on Dynamic scene dataset.

**Strengths:**

Strengths:
-The paper is well written with the objective clearly identified. It is structured well and has logically moving sub-section-wise explanations.
-Tackles a well-known problem in terms of generating novel view synthesis for dynamic scenes, but for semantics.
- Proposes a novel idea, that leverages optical flow to predict semantic labels for dynamic foreground pixels/regions.
- Evaluate and compare the model on the Dynamic Scene Dataset.

**Weaknesses:**

Weakness:
- Paper leverages optical flow output as one of the intermediate steps, but fails to discuss its shortcomings and how exactly do they handle occlusion and disocclusion related to both dynamic and static regions of the frame.
- For the most part of the paper, the authors only compare with two dynamic scene-based works, Considering other related works in dynamic scene reconstruction, Would be great to see comparative baseline results, with a few more of these models with semantic head.

**Questions:**

- Could Authors share some of the shortcomings (a few qualitative results) which may be due to imperfect flow prediction, which results in bad performance during inference?
- In Section 3.4: while calculating Semantic Consistency Constraint; Do we generate some sort of valid mask here to enforce the semantic consistency or is it done for all pixels, irrespective of occlusion or uncertainty?

---

> ### Author Response · Authors · 2023-11-19
> **Response to Reviewer 5VQj**
>
> # Shortcomings
> Thank you for your suggestion. We present the shortcoming of our model **in Figure 10 in the Appendix**. Due to the imperfect predictions of flows in the tracking setting, our model fails to estimate precise semantic labels of dynamic parts.
>
> # Occlusion and disocclusion problems
> To evaluate the performance of our model in the occlusion and disocclusion situations, as shown **in Section G.4 and Figure 7 in Appendix**, we add an occlusion region in both the static background and dynamic foreground in the frame #6, and remove the occlusion in the frames #5 and #7. Concretely, we manually occludes the both RGB colors and semantic labels of the region in the frame #6. As Figure 7 shows, the optical flow estimation method fails to estimate the object movement due to the occlusion. We train the model with the occluded image, wrong optical flow maps and semantic labels. The occlusion may provide wrong information to train the semantics, which leads to the incorrect semantic prediction in the occlusion part. However, since our model learns the semantics from flows, it successfully predicts the semantics in disocclusion parts by incorporating correct information among the flows in non-occluded regions.
>
> # More comparisons with other related works
> Thank you for your suggestion. We have tested other methods related to dynamic scene reconstruction **in Table 1 in the paper** and have a discussion about the performance of these methods in the **General Response**. Besides, we also conduct comparisons with the state-of-the-art video tracking method and the image segmentation method **in Table 2 and Figure 8 in the paper and Appendix.**
>
> # Semantic consistency constraint
> We conduct the constraint for all pixels. We visualize the pixel correspondence in consecutive frames **in Section G.2 and Figure 6 in the paper** and discuss the correspondence **in the General Response**.  Figure 6 shows that our model could generate the flows that map the similar parts of the dynamic foreground moving across time. We also discuss the occlusion problem **in Section G.4** and the uncertainty of noisy flows **in Table 13 and Section G.3 in the Appendix**.

---

### Official Review · Reviewer_xtsS · 2023-11-01

**Soundness:** 3 good
**Presentation:** 3 good
**Contribution:** 2 fair
**Rating:** 6
**Confidence:** 4

**Summary:**

The paper introduces learning a semantic field of the dynamic scene using NeRF given a monocular video. Given a sequence of input frames from a monocular video, precomputed optical flow, their Dynamic NeRF learns a semantic field so that it can render a semantic segmentation map at novel views. The method can be used for a couple of applications that output semantic field for unseen frames given partial frames.

**Strengths:**

- Better accuracy over baseline methods

  Compared to the two baselines, the method shows better accuracy on multiple tasks (scene completion, scene tracking, and semantic representation) in Table 1 and Table 2. Also it demonstrates better qualitative results in Fig. 3

- New applications

  The paper proposes interesting new applications, both dynamic scene tracking and completion that estimates semantic maps on unseen frames. (Fig. 1)

**Weaknesses:**

- Outdated baselines

  The paper compares their method with a couple of baselines (DynNeRF and MonoNeRF) but those are a bit limited. There are many other baselines for the dynamic NeRF task such as D-NeRF, RoDynRF, NSFF (Neural Scene Flow Field), etc. It would have been great if the paper provided accuracy on more baseline methods to make the comparison much fairer.

- A bit difficult to follow the equations (from Eq. (4) to Eq. (8))

  I am wondering if it's possible to put the mathematic notation from Eq. (4) to Eq. (8) into Fig. 2 for better understanding.

- Clarity

  Some parts of the paper have lack of clarity and make it hard to understand clearly. What is the meaning of '25%/50% semantic labels' in Fig. 3? I wonder if the paper can provide more details in the figure captions. How are the 25%/50% determined?

- Marginal accuracy improvement in Fig. 4

  The choice of low displacement seems not so critical for the accuracy gain. Maybe it would be good to have a justification or discussion on the result.

**Questions:**

- How much does the accuracy of the method depend on the off-the-shelf optical flow methods? Can it be critical?

---

> ### Author Response · Authors · 2023-11-19
> **Response to Reviewer xtsS**
>
> # Outdated baselines
> Thank you for your suggestion. We test the performance of NeRF [1] with time, D-NeRF [2], RoDynRF [3], NSFF [4] and discuss these methods **in Table 1 in the paper and in the General Response**. Besides, we also conduct comparisons with the state-of-the-art video tracking method and the image segmentation method **in the General Response** and **in Table 2 in the paper and Figure 8 in the Appendix.**
>
> # Difficult to follow the equations 4-8
> Thank you for your suggestion. We have revised **Figure 2 in the paper** to put the mathematic notations from equations 4-8 into the figure.
>
> # Clarity
> Thank you for your suggestion. In Figure 3, "25% semantic labels" in the tracking setting denotes that we use the frames #1, #2 and #3 to train the model and conduct semantic view synthesis on the frames #4-12. "25%" is determined by choosing 3 out of 12 frames to train the models in a scene. "50% semantic labels" in the completion setting denotes that we select the frames #1-3 and #10-12 to train the model and render semantic views in frames frames #4-9. "50%" is determined by choosing 6 out of 12 frames to train the models. We have added these details to the caption **in Figure 3 in the paper**.
>
> # Marginal accuracy improvement
> Thank you for your question. We agree that the results in Table 4 cannot distinguish $\boldsymbol{\Gamma}$ \& $\Delta \boldsymbol{\Gamma}$, $\Delta \boldsymbol{\Gamma}$ and $\boldsymbol{\Gamma}$. However, we demonstrate that flow displacements contribute to the semantic prediction performance according to the ablation study in Table 6 in the paper and our Semantic Flow has robustness with different flow displacements according to the Table 4. **In Figure 9 in the Appendix**, we visualize the semantic predictions by using $\boldsymbol{\Gamma}$ \& $\Delta \boldsymbol{\Gamma}$, $\Delta \boldsymbol{\Gamma}$ and $\boldsymbol{\Gamma}$. Although Table 4 shows quantitatively comparable results by using different displacements, Figure 9 presents that the semantic prediction of using $\boldsymbol{\Gamma}$ \& $\Delta \boldsymbol{\Gamma}$ is qualitatively better than using $\Delta \boldsymbol{\Gamma}$ or $\boldsymbol{\Gamma}$.
>
> # Relationship between the performance of semantic predictions and optical flows
> Thank you for your question. We test how much the performance of our model depends on the predicted flows from two perspectives: choosing different flow estimation methods and manually adding noise to the flow maps.
>
> Firstly, we choose RAFT [1] and FlowNet [2] to generate the optical maps and test the performance in the setting of semantic learning from multiple scenes. The results are presented **in the following table** and **in Table 12 in the Appendix**. According to the RAFT paper, RAFT outperforms FlowNet by a large margin in various optical flow estimation tasks. Therefore, the results show that there is about 10% performance drop in mIOU matrix by using FlowNet when jointly optimizing Jumping and Skating scenes, where the dynamic foregrounds are drastically changing and FlowNet fails to predict the correct flows. On the other hand, our model reaches comparable results when jointly optimizing Balloon1 and Balloon2 scenes where the foreground variations are relatively small and FlowNet successfuly predicts the correct flows of the foregrounds.
>
> | flow estimation method| year  | Balloon1      | Balloon2      | Jumping       | Skating       |
> |------------|------------|---------------|---------------|---------------|---------------|
> | RAFT [1]   | 2020| 0.919 / 0.844 | 0.967 / 0.839 | 0.936 / 0.733 | 0.970 / 0.716 |
> | FlowNet [2] |2015  | 0.926 / 0.855 | 0.974 / 0.766 | 0.921 / 0.636 | 0.965 / 0.591 |
>
> In addition, we manually add different percentages of noise to the predicted optical flows and test the performance of our model with noisy optical flows. The details are stated **in the Section G.3 in the Appendix** and the results are presented **in the following table** and listed **in Table 13 in the Appendix**. Our model could achieve comparable performance by using the optical flows with small percentages of noise (< 5%). Adding more than 10% noise to the predicted flows leads to a performance decrease in mIOU matrix, which is mainly because our model could not build accurate flow fields from noisy optical flows and hence learns inaccurate semantics from imprecise flows.
>
> | noise scale  | Balloon1      | Balloon2      | Jumping       | Skating       |
> |------------|---------------|---------------|---------------|---------------|
> | w/o. noise   | 0.919 / 0.844 | 0.967 / 0.839 | 0.936 / 0.733 | 0.970 / 0.716 |
> | 1% noise   | 0.923 / 0.844 | 0.963 / 0.820 | 0.937 / 0.715 | 0.970 / 0.707 |
> | 5% noise   | 0.924 / 0.808 | 0.964 / 0.813 | 0.937 / 0.656 | 0.969 / 0.714 |
> | 10 % noise | 0.922 / 0.777 | 0.964 / 0.666 | 0.936 / 0.690 | 0.969 / 0.670 |

---

> > ### Author Response · Authors · 2023-11-20
> >
> > References:
> >
> > [1] Teed, Zachary, and Jia Deng. "RAFT: Recurrent all-pairs field transforms for optical flow." In ECCV, 2020.
> >
> > [2] Dosovitskiy, Alexey, Philipp Fischer, Eddy Ilg, Philip Hausser, Caner Hazirbas, Vladimir Golkov, Patrick Van Der Smagt, Daniel Cremers, and Thomas Brox. "FlowNet: Learning optical flow with convolutional networks." In ICCV, pp. 2758-2766. 2015.

---

> > ### Comment · Reviewer_xtsS · 2023-11-22
> > **Response**
> >
> > Thank you so much for the responses! They are really helpful in understanding the paper!
> >
> > By the way, one minor question on the experiment using FlowNet. Why does FlowNet yield better results on Balloon1 and Balloon2 than RAFT? Does it mean that the optical flow signal might not be so critical for better segmentation accuracy? I would assume that flow quality near object boundaries would matter though.

---

> > > ### Author Response · Authors · 2023-11-22
> > >
> > > Thank you for your reply!
> > >
> > > We compare the flow prediction results from RAFT and FlowNet **in Figure 11 at the end of the Appendix**. In Figure 11, FlowNet fails to predict the foreground movements in the Jumping scene, which causes the performance drop in Table 12 in the Appendix. In the Balloon1 scene, although FlowNet predicts less accurate flow maps than RAFT, it successfully predicts the boundary of the foreground movement. The inaccuracy could be considered as a small scale of noise in the flow maps, by which our model achieves comparable performance or even about 1% improvement in the Balloon1 scene as shown in Table 13 in the Appendix.
> > >
> > > In conclusion,  inaccurate optical flow maps with a clear boundary of foreground movements may contribute equal or slightly better to the performance. However, wrong optical flow maps fail to provide information for semantic prediction and lead to a performance drop.

---

> ### Comment · Reviewer_xtsS · 2023-11-23
>
> Thank you for providing the flow map visualization.
>
> Then can I ask how adding noise (or using noisy GT) can improve the performance? As it's based on NeRF, using a better GT would contribute to better accuracy, unlike training-based approaches where noise could make models more robust by perturbing GT.

---

> ### Author Response · Authors · 2023-11-23
>
> Thank you for your response!
>
> Training Semantic Flow with the noisy optical flow maps leads to the noisy flow field predicted by $\Psi_{flow}$ in equation (3) in the paper. However, since Semantic Flow takes the flow information as input to predict semantic labels as shown in equations (6)-(9) in the paper, the noisy flows can be considered as an augmentation of flow information. In this way, although the predicted flow field is inaccurate compared to the ground truth flow map, it contributes to the semantic prediction, which is similar to adding Gaussian noise to the inputs for performance improvement in traditional training-based approaches.
>
> Again, we appreciate the reviewer’s continuous efforts to provide helpful advice and valuable input. If there is any further information or clarification you need from us, please feel free to let us know!

---

> > ### Comment · Reviewer_xtsS · 2023-11-23
> >
> > Thanks for the response. I updated my rating to marginally above the acceptance threshold. Clarity has been improved, compared with the initial draft. The authors' responses resolved most of my main concerns.

---

> > > ### Author Response · Authors · 2023-11-23
> > >
> > > Thank you so much for updating the score! We are very happy to address the reviewer's concerns!

---

### Author Response · Authors · 2023-11-19
**General Response (1/2)**

`This is the first part of the General Response.`

> **We sincerely thank all the reviewers for their thoughtful reviews. We have carefully revised the paper based on the reviews. We also highlight the changes in the paper with magenta and yellow colors.**

# Comparisons to more related works, the SOTA tracking method and the image segmentation method (Reviewer xtsS, Reviewer 5VQj and Reviewer QtRh)
Thank you for your question. The reason why we choose MonoNeRF [6] and DynNeRF [8] for comparison is that they are the most competitive methods in the semantic learning setting. We still agree that we should compare more baselines.
During the rebuttal phase, we re-implement a series of related works [1,2,3,4] and test the semantic prediction performance of these methods by adding a semantic head to each model. We present the results on 2 settings **in Table 1 in the paper**: semantic representation learning from multiple scenes and semantic adaption of novel scenes. We also provide the results **in the following table**. It is worth noting that since NSFF [1], NeRF [2] with time, D-NeRF [3] and RoDynRF [4] learn the semantics from positional embeddings, view directions and other information related to individual points, they lack motion information in the temporal dimension and hence severely overfit to the training frames. Besides, as D-NeRF studies the geometries and deformations of dynamic objects, it meets difficulty in handling complex dynamic scenes with various static backgrounds. Based on the performance in the table, we find that MonoNeRF and DynNeRF are still two of the most competitive methods in our setting.

| Total Acc / mIoU  | Balloon1      | Balloon2      | Jumping        | Skating        | Umbrella      | Playground    | Truck         |
|-------------------|---------------|---------------|----------------|----------------|---------------|---------------|---------------|
| NeRF [2]+time         | 0.780 / 0.529 | 0.455 / 0.211 | 0.913 / 0.564  | 0.857 / 0.277  | 0.953 / 0.763 | 0.919 / 0.416 | 0.920 / 0.433 |
| NSFF [1]              | 0.583 / 0.402 | 0.511 / 0.224 | 0.865 / 0.420  | 0.891 / 0.337  | 0.954 / 0.773 | 0.910 / 0.623 | 0.918 / 0.614 |
| RoDyNeRF  [4]         | 0.567 / 0.459 | 0.531 / 0.312 | 0.886 / 0.432  | 0.888 / 0.536  | 0.900 / 0.312 | 0.876 / 0.432 | 0.904 / 0.606 |
| D-NeRF [3]           | 0.267 / 0.079 | 0.381 / 0.076 | 0.267 / 0.135  | 0.305 / 0.136  | 0.700 / 0.221 | 0.605 / 0.323 | 0.780 / 0.501 |
| DynNeRF [8]           | 0.767 / 0.515 | 0.459 / 0.229 | 0.912 / 0.570  | 0.955 / 0.431  | 0.941 / 0.795 | 0.914 / 0.387 | 0.967 / 0.662 |
| MonoNeRF [6]         | 0.907 / 0.760 | **0.967** / 0.616 | 0.929 / 0.576  | **0.973** / 0.590  | 0.961 / 0.685 | 0.879 / 0.393 | 0.968 / 0.425 |
| Semantic Flow     | **0.919 / 0.844**  | **0.967 / 0.839** | **0.938  / 0.703** | 0.970  / **0.608** | **0.970 / 0.884** | **0.937 / 0.742** | **0.977 / 0.765** |


In addition, to compare with the state-of-the-art video tracking method, we apply the DeAOT [5] method to the RGB color rendering results from the MonoNeRF [6] method. We present the results **in the following table** and add the results to **Tables 2, 9, and 10 in the paper and Appendix.** Since DeAOT predicts semantics based on the rendered 2D images, it lacks the semantic understanding of 3D scenes and hence cannot hold semantic consistency across different views. As a result, our Semantic Flow outperforms DeAOT by reasoning the view-agnostic semantic label of each point in the 3D scene.
Besides, we also visualize the segmentation results obtained from Mask R-CNN [7] **in Section G.5 and Figure 8 in the Appendix.** Because the image segmentation method only conducts predictions on single images, it cannot hold the consistency of semantic labels in video frames. As Figure 8 shows, while Mask R-CNN aligns different semantic labels to the same person instances in different video frames, our model successfully predicts consistent semantic labels of the person instances in the entire video.

| method        | Total Acc (Completion) | Avg Acc (Completion) | mIoU (Completion) | Total Acc (Tracking) | Avg Acc (Tracking) | mIoU (Tracking) |
|---------------|------------------------|----------------------|-------------------|----------------------|--------------------|-----------------|
| DeAOT [5]        | 0.816                  | 0.600                  | 0.412             | 0.776                | 0.533              | 0.359           |
| DynNeRF [8]      | 0.934                  | 0.849                | 0.738             | 0.896                | 0.760               | 0.660            |
| MonoNeRF [6]     | 0.956                  | 0.891                | 0.786             | 0.935                | 0.818              | 0.716           |
| Semantic Flow | **0.961**                  | **0.901**               | **0.818**             | **0.942**                | **0.835**              | **0.767**          |

---

### Author Response · Authors · 2023-11-19
**General Response (2/2)**

`This is the second part of the General Response.`
# Discussion about semantic consistency of flows (Reviewer 5VQj and Reviewer QtRh)

Thank you for your insightful question. **In Figure 6 and Section G.2 in the Appendix**, we visualize the rendered images, estimated flow fields, semantic predictions and the point correspondence of dynamic foregrounds in 3 consecutive frames. It can be seen that our model could predict reasonable forward and backward flows. We agree that our semantic consistency constraint (Equation (16) in the paper) does not strictly force the estimated flows to map the the same specific parts across time. However, the visualization of correspondence shows that our model automatically generates the scene flows that map the relatively similar parts of the dynamic foreground.


References:

[1] Li, Zhengqi, Simon Niklaus, Noah Snavely, and Oliver Wang. "Neural scene flow fields for space-time view synthesis of dynamic scenes." In CVPR, pp. 6498-6508. 2021.

[2] Mildenhall, Ben, Pratul P. Srinivasan, Matthew Tancik, Jonathan T. Barron, Ravi Ramamoorthi, and Ren Ng. "NeRF: representing scenes as neural radiance fields for view synthesis." In ECCV, pp. 405-421. 2020.

[3] Pumarola, Albert, Enric Corona, Gerard Pons-Moll, and Francesc Moreno-Noguer. "D-NeRF: Neural radiance fields for dynamic scenes." In CVPR, pp. 10318-10327. 2021.

[4] Liu, Yu-Lun, Chen Gao, Andreas Meuleman, Hung-Yu Tseng, Ayush Saraf, Changil Kim, Yung-Yu Chuang, Johannes Kopf, and Jia-Bin Huang. "Robust dynamic radiance fields." In CVPR, pp. 13-23. 2023.

[5] Yang, Zongxin, and Yi Yang. "Decoupling features in hierarchical propagation for video object segmentation." In NIPS, pp. 36324-36336, 2022.

[6] Tian, Fengrui, Shaoyi Du, and Yueqi Duan. "MonoNeRF: Learning a generalizable dynamic radiance field from monocular videos." In ICCV, 2023.

[7] He, Kaiming, Georgia Gkioxari, Piotr Dollár, and Ross Girshick. "Mask R-CNN." In ICCV, pp. 2961-2969. 2017.

[8] Gao, Chen, Ayush Saraf, Johannes Kopf, and Jia-Bin Huang. "Dynamic view synthesis from dynamic monocular video." In ICCV, pp. 5712-5721. 2021.

---

### Meta-Review · Area_Chair_ZorG · 2023-12-11

**Metareview:**

The paper proposes a new problem of learning semantic fields so that it can render sementic maps at novel views. This has been tackled in static scenes, but this paper extends it to handle dyanmic scenes.

The reviewers' initial concerns were
- outdated competing methods
- missing important baselines based on running video object segmentations on RGB rendered images.
- some clarity issues.

The authors' rebuttal provided
- additional baselines (NeRF [1] with time, D-NeRF [2], RoDynRF [3], NSFF [4]),
- results with different flow networks, and
- applying DeAOT and Mask R-CNN to the rendered images from MonoNeRF.

After the discussion, Reviewer xtsS raised the score to "6: marginally above the acceptance threshold". Reviewer QtRh did not update the score but the AC reviewed the authors responses and believe that the raised concerns have been sufficiently addressed.

Based on the rebuttal and discussions, the AE thus recommends to accept.

**Justification For Why Not Higher Score:**

The improvement over the baseline is rather marginal.

**Justification For Why Not Lower Score:**

After the rebuttal, the initial conerns were properly addressed. The paper has sufficient merits for publication at ICLR.

---

### Decision · Program_Chairs · 2024-01-16

Accept (poster)